# Concept Mapping in Magnetism and Electrostatics: Core Concepts and Development over Time

**Christian M. Thurn** [1,*] , **Brigitte Hänger** [2] and **Tommi Kokkonen** [3]

[1]  Research on Learning and Instruction, Institute for Behavioral Sciences, ETH Zürich, Clausiusstr. 59, 8092 Zürich, Switzerland

[2]  Professur für Naturwissenschaftsdidaktik, FHNW, Hofackerstrasse 30, 4132 Muttenz, Switzerland; brigitte.haenger@fhnw.ch

[3]  Department of Physics, Faculty of Science, University of Helsinki, P.O. Box 64, 00014 Helsinki, Finland; tommi.kokkonen@helsinki.fi

\*  Correspondence: christian.thurn@ifv.gess.ethz.ch

**Abstract:** Conceptual change theories assume that knowledge structures grow during the learning process but also get reorganized. Yet, this reorganization process itself is hard to examine. By using concept maps, we examined the changes in students' knowledge structures and linked it to conceptual change theory. In a longitudinal study, thirty high-achieving students ($M$ = 14.41 years) drew concept maps at three timepoints across a teaching unit on magnetism and electrostatics. In total, 87 concept maps were analyzed using betweenness and PageRank centrality as well as a clustering algorithm. We also compared the students' concept maps to four expert maps on the topic. Besides a growth of the knowledge network, the results indicated a reorganization, with first a fragmentation during the unit, followed by an integration of knowledge at the end of the unit. Thus, our analysis revealed that the process of conceptual change on this topic was non-linear. Moreover, the terms used in the concept maps varied in their centrality, with more abstract terms being more central and thus more important for the structure of the map. We also suggest ideas for the usage of concept maps in class.

**Keywords:** concept mapping; science education; conceptual change; network analysis

## 1. Introduction

Children do not enter science education as tabulae rasae [1] but have acquired prior knowledge, for example, by prior exposure to the topic or from experiences in their everyday life. This prior knowledge often contradicts the target knowledge. Hence, learners' prior knowledge may hinder learning, as new knowledge cannot be integrated into the existing network of knowledge. The processes of acquiring new knowledge and restructuring the existing knowledge are termed conceptual change. Theories on conceptual change pose different hypotheses about the nature of students' prior knowledge and the processes of learning [2–5]. These various theories on conceptual change agree, insofar as they all assume, that conceptual change does not only require enrichment (student learns new "facts") but also structural changes and reorganization or changes in the underlying, implicit beliefs [2,3,6].

One of the central questions to the conceptual change research is the coherence (or integration) of students' knowledge. Research on science learning often pits novices' knowledge against experts' knowledge, which is thought to be not only richer but also more coherent and well-organized [7]. Learning is not only about collecting facts but also about acquiring a coherent, interconnected knowledge system that encompasses a large number of concepts [8]. The knowledge structure between novices, advanced students, and experts should show remarkable differences regarding the fragmentation and

integration of knowledge. However, few studies on conceptual change explicitly pay attention to the structural aspects of knowledge [9].

One promising method to investigate the structure of knowledge is concept mapping. Concept mapping has been known as an instruction as well as an assessment method since Novak [10] described it in the 1980s. It has been also used to investigate students' learning (see [8,11,12]). In this study, we use concept mapping to investigate students' learning across time in the light of conceptual change. Combined with network analytic measures, concept mapping can reveal interesting information on the process of knowledge restructuring that is assumed to happen in conceptual change. We apply such measures and investigate which measures best describe the learning process. Some exceptions notwithstanding, there are still few studies directly addressing the structural aspects of students' knowledge and especially its development across time. In this study, we will also explore which concepts are central for students' understanding and for conceptual change to occur. As misconceptions often hinder the learning process, we also aim to identify the most common misconceptions and their development. To infer about the learning processes, we will also compare the student concept maps against concept maps from experts.

*1.1. Conceptual Change Theories*

Learning of scientific concepts is often examined from the perspective of conceptual change. This research field evolved largely out of interest on how students' prior knowledge hinders learning and why misconceptions persist even after ample instruction. The term conceptual change denotes the many kinds of transformation processes, where students' initial knowledge transforms towards scientific knowledge. Yet, conceptual change theories differ with regards to the nature of the knowledge and its reorganization. Students' knowledge has been described, for example, in terms of ontologies [6], implicit theoretical beliefs [13], and phenomenological primitives (p-prims) [14], that are kinds of naïve causal intuitions.

The different views on conceptual change more or less agree that learning entails changes at multiple levels of the knowledge system [2,6,14]. Learners might have gaps in their knowledge whereby they would need to elaborate their knowledge and simply learn new facts [2,6]. They might also have misconceptions (such as "there is oxygen on the moon"), which need to be refuted. Conceptual change in this level may consist of rather simple assimilation of new knowledge into the existing knowledge structure and elaboration of the prior knowledge [2,6]. However, much of the conceptual change literature focuses on changes in the underlying, often implicit knowledge structures and restructuring of the knowledge [2,6]. These types of changes are often thought to be more difficult and time-consuming. Hence, conceptual change comes in degrees and types—some simpler to achieve than others.

Alexander [15] speculates that in the beginning, during the acclimation phase, learners first have to familiarize themselves with the new domain and learning is highly fragmented. At this stage, learners' knowledge will be piecemeal, meaning that various concepts exist beside each other and are not well linked. Thus, misconceptions can often occur. After some time and studying the topic, the learner will have established some competence in the domain and can generate general principles that help to integrate the fragmented knowledge. Interconnections among the knowledge fragments are drawn and misconceptions are corrected. Finally, the learner can master a content area and become an expert if they have constructed an integrated knowledge base. At this stage, the expert might also generate new knowledge, but few people are reaching this point. Experts differ from novices in their knowledge organization and represent knowledge with different levels of explanatory depth [1]. Whereas experts focus on core properties or defining features, novices tend to orient at peripheral properties or surface features [3]. During the process of conceptual change, learners become to represent knowledge more on abstract and less on situated or context-based levels.

Especially in physics, the target knowledge can be described in terms of vast interconnected network structures [16]. Learning such knowledge takes a long time during which misconceptions and fragmentation can occur. The theories of conceptual change acknowledge that students' knowledge cannot be inspected in isolation—that is, learning involves changes in the interconnected elements

of the knowledge structure [17]. Often the accounts of conceptual change are ambiguous about the relational aspects of conceptual change [17–19]. In addition, structural features other than coherence are not widely discussed. Recently, however, the role of relations and structural aspects in learning conceptual knowledge have been increasingly emphasized [8]. Specifically, it has been emphasized that understanding of concepts is based on the relations that concepts bear to other concepts [8]. Taken together, this lends credibility to viewing the knowledge as a network on which one can apply network analysis methods to explore conceptual change processes. Towards this end, concept mapping is a promising method that can reveal the global structure of the learners' knowledge network.

The more disciplinary oriented research (e.g., physics education research) has specifically concentrated on the specific conceptions (sometimes called "naïve", "false", or "mis-" conceptions) students have on various topics [20]. Especially the cognitively oriented conceptual change research, on the other hand, has also aimed at revealing the general conceptual change processes [20]. In the context of physics, conceptual change has been investigated widely and for several topics, such as light, heat, electric current, the shape of the earth, friction, gravitation, or buoyancy [21–25].

In this article, we assume that the relational aspects of concepts are central to understanding. This way we seek to extend and complement the previous accounts of conceptual change by focusing specifically on the structural analysis of students' knowledge and on the development of relational knowledge across time. We are focusing on magnetism and electrostatics and specifically on the concept of field (see Section 2.2 for details). Learning of magnetism and electrostatics provides an excellent basis for investigating transfer and is core for understanding sophisticated topics such as electromagnetic phenomena and applications, the Lorentz force and electromagnetic induction, electromotors or generators. Although there are numerous similarities between magnets and electric charges, there are also important differences, such as that electric charges can exist as monopoles, whereas magnets always come in dipoles. The combination of Electricity and Magnetism (E&M) is an obligatory topic in secondary school and many undergraduate science courses at university treat this topic. Despite its importance and prominence, the basic concepts in this topic are poorly understood [26].

There are concepts in the area of electrostatics, such as field line depictions, which can be understood only from a conceptual perspective. Field line depictions can be drawn for magnets as well as for electric charges and for various objects, such as bar magnets, coils with an electric current, or charged spheres from a generator. Field line representations are used very often in schoolbooks to demonstrate special characteristics of a certain object or technical gadget, such as a capacitor or electric motor. Yet, children face problems with these representations. Previous research has identified various difficulties students have—students do not have a clear understanding about what fields are or how electric and magnetic fields differ. Students may also confuse different fields, such as magnetic and electrostatic fields, and think of the field as a flow from negative charges towards positive ones, which ends at a certain point [27]. They also confuse field lines with the trajectory of a particle—that is, they think that a charged particle would travel along field lines [28]. The correct conception is that field lines indicate the direction of the force, (and hence, the acceleration of a particle interacting with the field) not the velocity. In addition, students sometimes fail to adhere to Newton's third law and the symmetry of Coulomb's law by thinking that a larger charge exerts a larger force [29]. Thus, the examination of misconceptions can yield valuable insights on the problems students have when learning this topic and teachers could address them in their teaching.

### 1.2. Concept Mapping

Assuming that physics knowledge is organized in a network-like structure and that learning entails elaboration and restructuring the network, concept maps offer a natural method of investigation [30,31]. Since their invention, concept maps have been used in science education as tools for supporting students' learning of the structural nature of physics knowledge but also as tools for assessment and evaluation of learning [8]. The method of concept mapping emerged to leverage the understanding of the process of conceptual change in science [32], growing out of theories describing cognitive structures

recognizing the interrelatedness of concepts as an essential property of knowledge [33]. That is, the assumption is that the meaning of concept is (at least partly) determined by its relations to other concepts [8,33]—making suitably constructed concept maps an adept tool for learning and assessment.

In this study, concept maps are used for research purposes—that is, as tools for assessment and evaluation. As such, concept maps are more related to declarative than to procedural knowledge [33]. That is, the maps are more related to conceptual knowledge than to knowledge about how the concepts are used or applied in problem solving. By the virtue of their nature, concept maps enable researchers to investigate the structural aspects of knowledge. For example, the use of more links and nodes or the development of a more coherent map over time can be regarded as a sign of a learning process. By requiring students to arrange a given set of terms and to justify the links between them, the organization and possible hierarchies of knowledge can also be inferred. Researchers have also used concept mapping to compare knowledge structures between novices and experts. For example, Koponen and Pehkonen [30] found that expert maps differ in the topology of the map as well as in their hierarchical structure from concept maps created by novices.

There is a lot of variation regarding the implementation of concept mapping related to the actual task, constraints, and contents [33]. Several methods are possible, ranging from free-style mapping of terms through providing lists of concepts and links to even providing a skeletal structure of the map. In the present study, the participants constructed concept maps from a list of given terms. The students were, however, free to arrange the terms as they liked—that is, there was no prescribed structure for the map (see Section 2.2 for more details).

When concept mapping is used repeatedly over time, conclusions concerning the development of knowledge can be drawn. How knowledge develops for a group of learners can reveal information about core concepts and misconceptions that may hinder the learning progress at a certain stage and which teachers can then approach in the classroom [34,35]. The evaluation methods of concept maps range from qualitative (e.g., visual inspection of structure or complexity) to quantitative methods (e.g., applying graph theoretical measures to the maps). Many different scoring schemes have been developed that may take not only the content (correct and incorrect conceptions, abstractness of terms) but also the structure into account (hierarchies, chain structures, star structures). Yet, these schemes have also been criticized as lacking internal consistency and validity [36,37]. Besides such scoring schemes, quantitative methods for evaluation based on network theory have evolved in recent years (e.g., [38]). These methods allow the identification and quantification, related especially to the global structure of the maps, to be evaluated beyond the qualitative inspection. Similar to the qualitative inspection of terms with different levels of abstractness in the scoring schemes, different types of terms can also be evaluated with network analytic methods to shed light on important terms that are central to mastering a topic. Moreover, in the context of science education, where conceptual change is core to understanding certain topics [5], concept mapping can yield valuable information for teachers by allowing them to see common elements that their students did not yet fully understand or that are prone to misconceptions.

*1.3. The Present Research*

This study investigates conceptual change by using the method of concept mapping across three timepoints: at the beginning of a topic, at an intermediate timepoint, and at the end of the topic. This allows us to examine key terms and structures that characterize every learning stage, as well as key misconceptions that may hinder progress. Furthermore, we link the development of the concept maps with conceptual change theories on reorganization of knowledge. We also investigate to what extent basic network analytic measures, such as centrality measures, depict such a conceptual change. Besides the longitudinal perspective, we explore how the student maps evolve compared to concept maps from experts in the topic.

The concept maps on the topic of magnetism and electrostatics constructed by a group of high-achieving secondary school students allow us to address the following research questions:

(a)   What are the central terms and common misconceptions across time, and how does terms' centrality differ between the timepoints?

(b)   What are the underlying structural properties of the concept maps across the learning process?

(c)   Are there differences regarding different types of words used in the concept map?

(d)   How do the student concept maps compare to the expert maps?

## 2. Materials and Methods

This study is based on a pre-registered study (https://osf.io/xkem5/), for which we were using concept maps to identify differences between students with high and low prior knowledge. However, the current study has to be considered explorative, as it does not approach any of the hypotheses mentioned in the preregistration.

### 2.1. Participants

Data on knowledge about magnetism was obtained from the Swiss MINT Study of ETH Zürich (see [39]), in which children receive inquiry-based curricula on physics topics in primary school. Participants were chosen on the criterion of having high prior knowledge in the topic of magnetism, defined by completing 15 lessons on this topic beforehand on which they scored above average in a posttest. These high-achieving students were invited to attend an advanced course on magnetism and electrostatics. Thirty students with a mean age of 14.41 years ($SD = 1.23$; 43.3% female) took part in this course.

Students who came into consideration were approached by mail, and the students as well as their parents had to give informed consent before taking part in this study. The whole project, of which this research is a part of, was approved by the Ethics Committee of ETH Zürich (EK 2019-N-35 on 17 April 2019).

### 2.2. Procedure

The course on magnetism and electrostatics consisted of three sessions and introduced the concepts of magnetic and electric fields as well as the gravitational field. Magnetic, electric, and gravitational forces are puzzling for students because of their property to act at a distance. The problem of the action at a distance is solved with the introduction of a force field. Thus, the aim of this unit was to show students how to use the concept of force fields as a mental tool in order to deal with the idea of action at a distance and to explain various magnetic and electric phenomena. Consequently, the unit often made use of field line depictions as supportive methods to enable the use of force fields as mental tools. The unit consisted of five different topics: (a) what is a field, (b) the direction of fields, (c) the strength of fields, (d) attraction and repulsion, and (e) the gravitational field, which was used as a transfer topic.

The instruction material made use of inquiry-based cognitively activating methods [40]. Physics instruction still follows rather traditional procedures despite ample evidence that it often is inefficient for instance in lifting students' prior conceptions. Cognitively activating instruction is a set of student-centered teaching methods focusing on conceptual understanding. It is intended to help take prior knowledge into account, change existing knowledge, and construct conceptual knowledge [25,40].

Whereas electricity, magnetism, and gravitation are conventionally treated in sequential order at school, in this unit the students were taught the topics in parallel using contrasting methods in order to emphasize the core concept of a force field, to facilitate transfer between the three topics, and integration of knowledge. Such contrasted comparisons are intended to help to differentiate superficially similar but differing concepts or phenomena by directly juxtaposing them and supporting the recognition of the similarities and differences between the concepts or phenomena. Whereas contrasted comparisons require more learning time and effort in the beginning, they help students eventually to carve out similarities and differences between the contrasted topics [41]. The material used visual clues to delimit the electric, magnetic, and gravitational field and to make clear for every experiment which field is actually associated with it.

Besides various experiments on the visualization of magnetic and electric fields as well as on attraction and repulsion, the teaching material contained self-explanations, texts, and thought experiments, and aimed at enabling the students to learn how to interpret graphical representations of field lines. Self-explanations are found to be an effective method in enhancing students understanding by guiding students to consciously reflect upon the teaching material [42,43]. The teaching material also contained a glossary for unknown terms that the students wanted to look up.

All students were instructed by the same teacher—the first author—in a facility at his research institution. As the course had to take place outside of school hours and appointments had to be made individually with each child, the course was carried out in small groups of one to six children per session. Generally, children were invited to come on three subsequent weeks on the same day, but due to individual constraints, the time between sessions also varied, with a mean of 7.63 days (*SD* = 8.92 days) from timepoint one to two and a mean of 6.96 days (*SD* = 5.79 days) between timepoint two and three.

After every unit, students constructed a concept map using the software CmapTools [44]. Here, the concept mapping was used primarily as an assessment tool for research purposes. That is, the concept maps were not part of the instructional unit *per se*. Constructing these concept maps may have affected students' learning, but the maps were not specifically used as supportive learning tools. That is, students were not instructed in which kind of map was a desirable one, given a skeletal structure, or asked to reflect upon the map.

The method of concept mapping has to be practiced carefully beforehand to make sure that everyone understands and applies the method correctly and to prevent any bias from difficulties with the method. Thus, we introduced concept mapping at the beginning of the first unit as an assessment tool to make knowledge visible. We explained, that in contrast to mind maps, there is not necessarily a single central concept or a certain hierarchy, but that concept maps reveal how several terms relate to each other. The students then practiced concept mapping with CmapTools, first by completing an incomplete concept map on the topic of sound and instruments to familiarize with the structure and the principle of justifying every link between concepts. Second, they also had to construct a concept map on their own on the topic of floating and sinking to get an idea how to structure terms in the concept map and how to use the software. The teacher also made the students aware that there is not a single solution, but that every concept map is correct in its own sense. The teacher also mentioned that the links in these concept maps can be drawn in any direction and the direction will not be relevant. At the end of the introduction, the teacher made the students aware that each student's concept map looked different. The teacher then prompted a metacognitive question—what can concept maps assess differently compared to typical exam questions? The teacher also made the students aware that they can use concept mapping outside of the course for summarizing a topic.

After each of the three units, students constructed a concept map with 35 terms provided on the topic of Magnetism and Electrostatics. See Figure 1 for an example of a concept map and the provided terms. On visual inspection, there are some unconnected nodes and the grammatical expression on the relations sometimes lack sophistication. The terms "Ferromagnetism" and "Electrostatics" have been arranged by the student below the category "I don't know this yet", meaning that they could not yet assign these two terms. Nevertheless, we regard this as a good concept map, stating the characteristics that a magnetic field is caused by electric current, an electric field by charges, and a gravitational field by mass. Whereas visual inspection is important to characterize the context of the nodes and to verify the interpretations, using network analysis helps to avoid subjective rating.

After each unit, students had 25 min to draw the concept map. In all cases, they had to construct a new map and could not access previous ones. They did not receive a root node or root question. Instead, the task was to depict how the different terms and depictions relate to each other. Altogether, the 30 students drew 87 concept maps (three students did not complete all three timepoints) which constitute the sample of concept maps that were analyzed in this study. Specific characteristics of the concept maps used in this study were that the students could link terms that they did not know with a specific node labelled "I don't know this yet" or with a node labelled "this doesn't fit". Moreover, nine

nodes consisted of visual depictions of field lines—magnets, electric charges, and cables that attract or repel each other, as well as the gravitational field of the earth. We used the field line depictions from the English Wikipedia [45]. See Table 1 for an overview of these pictorial nodes and their labels that are used throughout the text. All the pictures contained field line visualizations that work as mental tools. Normally, electric and magnetic fields are invisible and have to be uncovered with certain materials, such as iron powder, small compasses or nails, or ferrofluid. Using such depictions in concept maps enables the representation of complex information, which cannot be described in one or two terms, which is the usual element of a concept map. Depictions are one way to put in a lot of information and background context in one node.

We also classified each verbal node into one of six categories. Terms that were graspable, real-world objects we classified as **"Objects"**, with "Magnet", "Compass", "Cobalt", "Iron", "Nickel", and "Metal" falling into this category. Terms that in contrast depicted mental tools that are not really observable but help to understand a physical observation—which the concept of the field characterizes—fell into the category **"Models"**, with "Magnetic Field", "Electric Field", "Geomagnetic Field", "Gravitational Field", "Field", "Field Lines", "Direction of Force", "Dipole", "Monopole", "Magnetic North Pole", and "Magnetic South Pole" falling into this category.

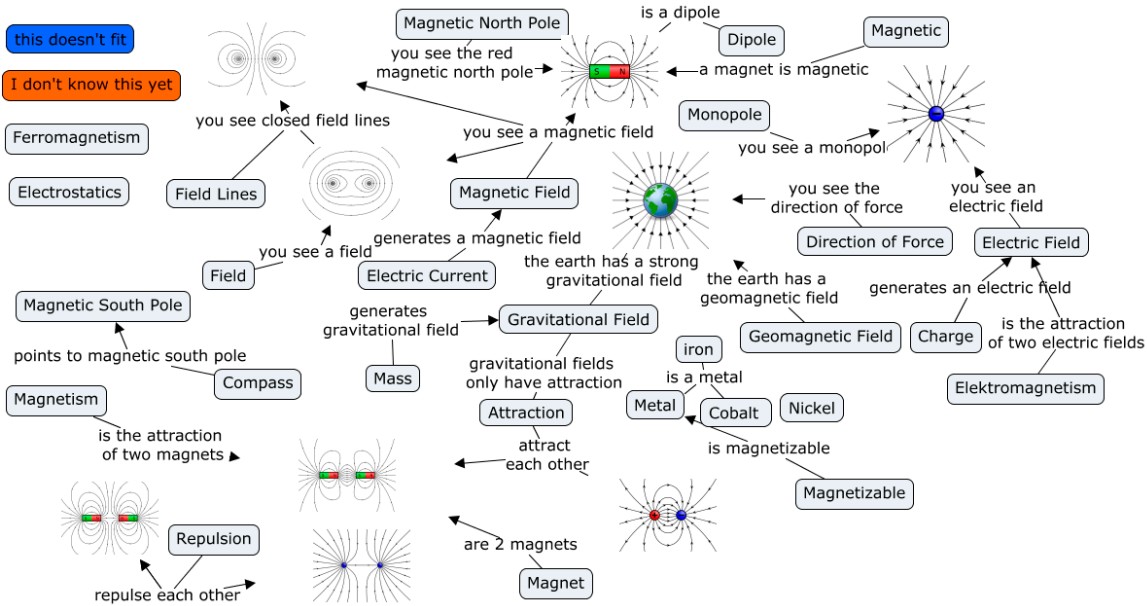

**Figure 1.** Concept map of a 14 year old male student at timepoint three. In blue and orange, you see the special nodes for unfitting/unknown items (translated from German).

Terms that were related to the topic of magnetism or electrostatics but were not clear physical models related to the field nor graspable objects were categorized separately. Terms that described quantities were categorized as **"Physical Quantities"**—"Charge", "Electric Current", and "Mass" fell in this category. The observable behaviors of "Attraction" and "Repulsion" were classified as **"Phenomena"**, and "Magnetic" and "Magnetizable" were classified as **"Attributes"**. The terms "Magnetism", "Ferromagnetism", "Electrostatics", and "Electromagnetism" were assumed to be terms of a higher hierarchy level and were categorized as **"Topics"**. Together with the depictions, we had all the nodes classified into seven categories (Depictions, Objects, Models, Physical Quantities, Phenomena, Attributes, and Topics) that we used to investigate conceptual differences between categories of terms.

**Table 1.** Pictorial nodes that were used in the concept maps and labels that are used throughout the article. Pictures from VectorFieldPlot (see [45]).

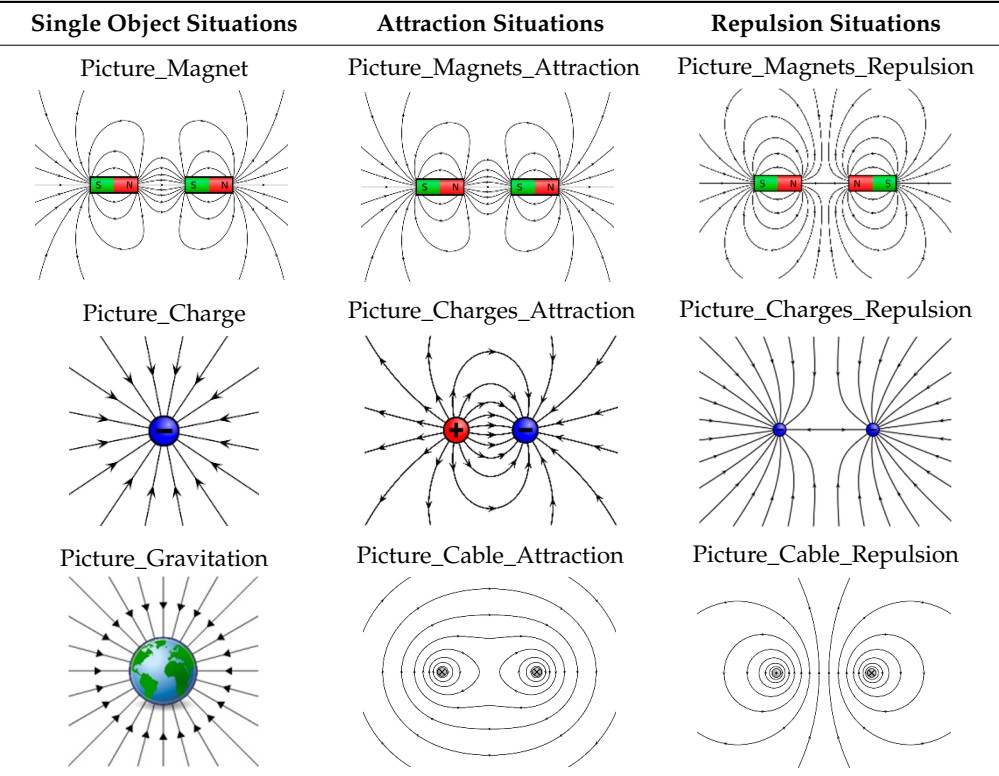

To investigate misconceptions, one rater—the first author—rated every link of each concept map in terms of misconceptions. Being rather conservative in this rating, only connections that clearly expressed conceptions that differed from the current state of scientific knowledge were coded as misconceptions. For example, the connection from the Geomagnetic Field to the Gravitational Field together with some label stating "this is a", "equal", or "depiction" was coded as a misconception, as the geomagnetic field is not the same as the gravitational field. Another example would be the connection from Charge to Magnetic North Pole together with the label "the charge determines the direction of the pole", which would be a misconception as magnetic poles do not have a charge.

In order to investigate expert knowledge, four experts constructed concept maps—the first author, the second author who holds a doctoral degree in physics, as well as one secondary school teacher from physics, and a chemist constructed concept maps independently from each other.

### 2.3. Statistical Analyses

A concept map can be regarded as a network consisting of nodes (also called vertices) and links (also called edges) between them, which enables the application of graph theory on concept maps. At first, we applied standard graph theoretic measures to get an overview of the properties of the concept maps. As the direction of links was explicitly mentioned to be irrelevant, we analyzed all maps as undirected graphs. To describe surface features of the concept maps, we counted the number of links and nodes at each timepoint, the density (which is the ratio of the number of existing links to the number of possible links), the mean distance (which is the average length of all shortest paths between nodes in the graph), and the diameter (which is the longest path between two nodes in the network). Not all children connected unused nodes to either the specific node labelled "I don't know this yet" or "this doesn't fit" but left them unconnected. We aggregated these three options to count the number of unknown nodes and analyzed them separately. Furthermore, we removed nodes that were added individually by some students, to ensure comparability of the concept maps across students.

Regarding the centrality of certain nodes, several measures exist, from counting the number of edges per node (degree centrality), over the average shortest path length to all other nodes (closeness centrality) to the number of paths that cross through a certain node (betweenness centrality). Each centrality measure answers a slightly different question. Using degree centrality (or other measures of radial centrality), we would suppose that only immediate neighbors of a node would be considered when this node was inserted in the concept map. Structure of the overall map would not play a role at all. Regarding the process of constructing a concept map, this assumption is probably too strict. Using closeness centrality, one would suppose that every shortest path of every node would be considered when this node was inserted in the concept map—an assumption that is also hardly justifiable. Yet, we assumed, that besides the directly adjacent neighbors also the paths that cross a certain node have been considered in the construction of the concept map and inform about the centrality of it. Thus, to combine the number of adjacent nodes as well as some structural properties of a node, we used betweenness centrality. Moreover, as a comparison measure, we applied PageRank centrality [46] that takes the number of (incoming) edges but also the importance of adjacent nodes sending these edges into account. In contrast to the aforementioned degree and betweenness centrality indices, PageRank centrality weights each edge differently according to the importance the node it emerges from. As PageRank centrality was developed for directed graphs but we used undirected graphs, we split each undirected edge into two directed ones for calculating it.

To investigate the underlying structure or common links across all students, we constructed aggregated graphs from all student maps. We aggregated the maps across all timepoints, but also separately for each timepoint. This aggregation enables researchers to interpret common pathways and structural elements that describe core features of the concept maps at each timepoint. The most common structure can be visualized when rare edges are left out. Thus, to be able to interpret the maps, we pruned the aggregated concept maps until only the most common edges remained. To identify homogeneous subgroups of terms in the concept maps, which reveal major topics and have similar characteristics, we used clustering of the concept maps. We applied a clustering algorithm based on Optimal Community Structure [47] to all student maps separately.

We used R for all analyses, with the following packages in alphabetical order: afex, CINNA, centiserve, comato, corrplot, dplyr, emmeans, ggplot2, igraph, lmerTest, magrittr, proxy, qgraph, rcartocolor, and readxl [48–62]. The R script and data is available on Open Science Framework (https://osf.io/behft/). We used an alpha level of 0.05 for all statistical tests.

## 3. Results

Regarding the results, we started with analyses on the global level of the maps, using summary statistics such as the number of edges, density, or diameter of the map across time to reveal structural changes. After that, we focused on specific terms that are central, indicated by their betweenness centrality as well as misconceptions that occur often to shed light on research question (a) *What are the central terms and common misconceptions across time, and how does terms' centrality differ between the timepoints?* Afterwards we show the results for the aggregated maps and the clustering algorithm, revealing the core structure that is typical for most students. This aims at research question (b) *What are the underlying structural properties of the concept maps across the learning process?* Then we group the terms into the aforementioned seven categories to tackle research question (c) *Are there differences regarding different types of words used in the concept map?*

Finally, we compare the students' results to four concept maps from experts. These results aim to answer research question (d) *How do the student concept maps compare to the expert maps?*

### 3.1. Development across Time

Regarding basic descriptive statistics of the graphs, the number of used concepts increased across time ($t_1$ = 24.9, $t_2$ = 27.1, $t_3$ = 30.4). We applied within-subjects ANOVAs with Type III sum of squares and a Greenhouse–Geisser correction of the degrees of freedom for violations of sphericity.

The ANOVAs revealed that the number of nodes increased significantly ($F(1.76, 47.41) = 17.92$, $p < 0.001$, $\eta_G^2 = 0.10$), as well as the number of edges ($t_1 = 25.1$, $t_2 = 26.9$, $t_3 = 32.5$, $F(1.91, 51.69) = 13.58$, $p < 0.001$, $\eta_G^2 = 0.10$). Thus, the concept maps seemed to grow larger across time, with more nodes getting integrated, and more links drawn. The density decreased ($t_1 = 0.089$, $t_2 = 0.082$, $t_3 = 0.077$), but not significantly ($F(1.82, 49.26) = 2.65$, $p = 0.09$, $\eta_G^2 = 0.03$). The mean distance decreased from timepoint one to two, to then increase again to timepoint three ($t_1 = 3.13$, $t_2 = 3.05$, $t_3 = 3.36$, $F(1.77, 47.87) = 1.72$, $p = 0.19$, $\eta_G^2 = 0.02$) as well as the diameter, but both not significantly ($t_1 = 6.63$, $t_2 = 6.41$, $t_3 = 7.25$, $F(1.77, 47.76) = 1.75$, $p = 0.19$, $\eta_G^2 = 0.02$). The decrease of these two measures from timepoint one to timepoint two and their recurring increase from timepoint two to timepoint three can be explained in two ways. Either one large component emerged at timepoint two, that had a lot of connections and thus shorter distances and a smaller diameter. Alternatively, timepoint two could be characterized by several smaller components, which would then show smaller diameters, and which would only get integrated at timepoint three. We investigated this pattern further (see below). A table with standard deviations, sample sizes, and confidence intervals for all above described statistics can be found in Appendix A Table A1.

### 3.1.1. Special Nodes and Landmarks

Betweenness centrality yields information about those nodes that are highly connected and build a structurally strong core of the concept map. Thus, we used this measure as a compromise between considering only directly adjacent links and considering the structure of the whole net for each node. We compared it to degree centrality and PageRank centrality indices and calculated Spearman rank-correlations between these indices for each timepoint. For timepoint one, betweenness correlated highly with degree ($r_s = 0.85$, $p < 0.001$) and PageRank centrality ($r_s = 0.86$, $p < 0.001$), as well as degree and PageRank correlated strongly ($r_s = 0.89$, $p < 0.001$). For timepoint two, the correlation between degree and PageRank was even higher ($r_s = 0.96$, $p < 0.001$) and remained comparably high for betweenness and degree ($r_s = 0.84$, $p < 0.001$) as well as for betweenness and PageRank ($r_s = 0.88$, $p < 0.001$). At timepoint three, all centrality indices—betweenness and degree ($r_s = 0.94$, $p < 0.001$), betweenness and PageRank ($r_s = 0.90$, $p < 0.001$), degree and PageRank ($r_s = 0.93$, $p < 0.001$)—again correlated highly. This indicated that the centrality indices agreed mostly in ranking the nodes based on different centrality measures.

Table 2 shows the most central terms or landmarks at each timepoint. As centrality indices strive to identify the most influential nodes, they are less suited for the nodes with low centrality and do not necessarily express a meaningful order of such nodes with lower centrality because of a lack of sensitivity. Thus, we focus on the centrality statistics for the first five nodes at each timepoint. The interested reader can find a table of the centrality of all nodes in Appendix A Table A2. Note also, that the indices can just reveal an order of important nodes, and that their absolute values do not inform about centrality differences on an interval scale.

**Table 2.** Most central nodes indicated by betweenness (B). Degree (D) and PageRank (PR) given for comparison. Terms that are most central in more than one timepoint are printed in bold.

| Timepoint 1 | | | | Timepoint 2 | | | | Timepoint 3 | | | |
|---|---|---|---|---|---|---|---|---|---|---|---|
| B | D | PR | Node | B | D | PR | Node | B | D | PR | Node |
| 0.26 | 3.14 | 0.058 | **Magne-tism** | 0.18 | 2.76 | 0.044 | **Magne-tism** | 0.2 | 3.4 | 0.042 | **Field** |
| 0.19 | 3.32 | 0.061 | **Magnet** | 0.17 | 3.00 | 0.049 | Ferromag-netism | 0.19 | 3.74 | 0.053 | **Magnet** |
| 0.17 | 2.71 | 0.053 | **Field** | 0.16 | 3.7 | 0.061 | **Magnet** | 0.16 | 2.92 | 0.040 | **Magnetic Field** |
| 0.15 | 2.54 | 0.048 | **Magnetic Field** | 0.14 | 3.29 | 0.053 | Electromag-netism | 0.15 | 3.21 | 0.043 | **Magne-tism** |
| 0.13 | 2.79 | 0.054 | Magnetic | 0.12 | 2.38 | 0.042 | **Magnetic Field** | 0.12 | 2.85 | 0.042 | Attraction |

At timepoint one, central nodes appear to be those that are related to (ferro-)magnetism, such as "Magnetism", "Magnet", and "Magnetic Field". Furthermore, the concept of "Field" shows a high centrality. In line with findings of [63], at timepoint two and three central nodes are found mostly to be

abstract, general, and advanced concepts, such as "Electromagnetism" and "Ferromagnetism" or the concept of the "Field". "Magnet", "Magnetism", and "Magnetic Field" are terms that remain central landmarks across all timepoints, revealing a rather strong stability of the concept maps regarding the most central terms.

Figure 2 shows for the terms that were most central at timepoint three how their ranks in betweenness centrality changed across time. We used ranks instead of actual values, as their absolute values are not comparable between timepoints on an interval scale. The centrality of the concept "Field" dropped a little to timepoint two, to then become the most central concept at timepoint three. This reveals that the concept of the field was strongly integrated at timepoint three, with many other links going through this central concept.

Regarding the unknown terms, their number decreased across time ($t_1$ = 8.98, $t_2$ = 7.58, $t_3$ = 4.35, $F(2,117)$ = 8.03, $p < 0.001$, $\eta^2$ = 0.12). We investigated the most frequent terms that students did not integrate in their concept maps. At timepoint one, the most frequent terms were "Dipole" ($n = 23$), "Monopole" ($n = 22$), and "Mass" ($n = 22$), so rather concrete terms that were related to the characteristics of magnets and charges and were only introduced during the lesson. At timepoint two, the three most frequent unknown terms were "Ferromagnetism" ($n = 24$), "Mass" ($n = 22$), and "Electrostatics" ($n = 17$), revealing more concepts that were not known before the lessons. At timepoint three, the most frequent unknown terms were again such concepts: "Ferromagnetism" ($n = 15$), "Electrostatics" ($n = 13$), and one pictorial node ("Picture_Cable_Repulsion", $n = 8$). Thus, after the lessons, 50% of the students still did not use the terms "Ferromagnetism" and "Electrostatics" in their concept maps. This result is not surprising, when one considers that the teaching unit did not explicitly introduce the term "Ferromagnetism" compared to the other magnetic phenomena of Dia- and Paramagnetism. As well, "Electrostatics" was a rarely used term, compared to "Electric Field", which was used more often.

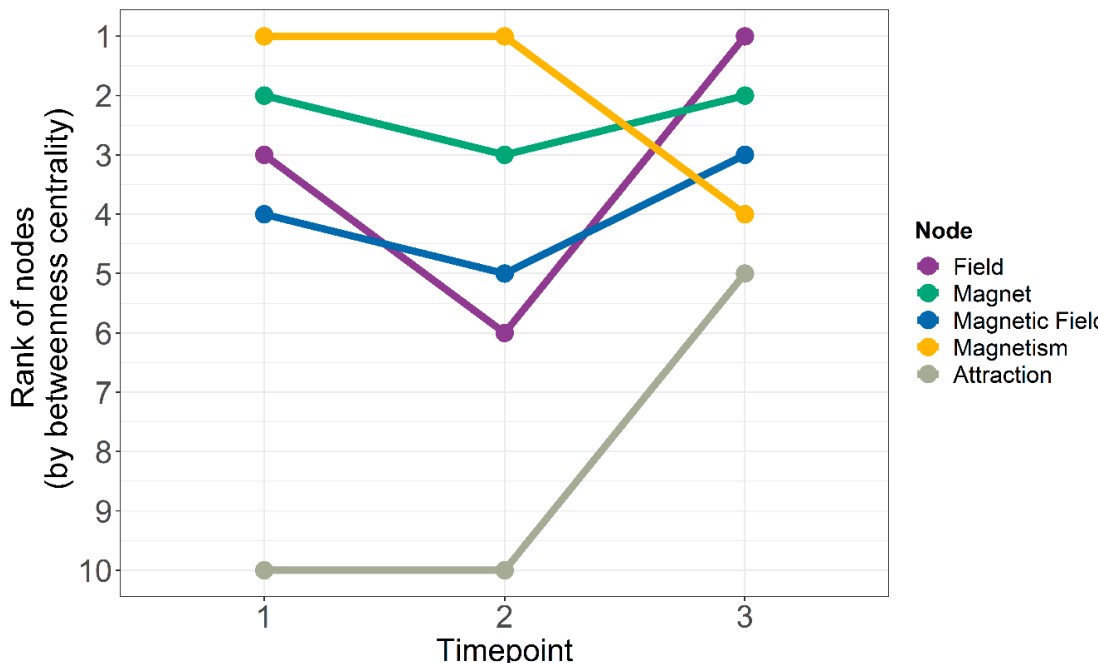

**Figure 2.** Ranks of betweenness centrality across time for central nodes at timepoint three.

### 3.1.2. Misconceptions

About 10%–12% of the edges relative to the total number of edges were coded as misconceptions at each timepoint ($t_1$: 162 misconceptions, 10.93%; $t_2$: 181 misconceptions, 11.85%; $t_3$: 216 misconceptions, 12.02%). As the coding of misconceptions could only happen on the level of edges, the misconceptions always involved the relation of two nodes. For example, the misconception that magnetic poles are

charged involves the concept of a magnetic pole as well as the concept of charges. At timepoint one, the term "Magnetic" was involved in most misconceptions in the thirty student maps (29 misconceptions). This term was often used in connection with the three ferromagnetic elements Nickel, Cobalt, and Iron. This is wrong, as these elements are not magnetic by themselves, but as soft magnetic materials they only are magnetized when exposed to a magnetic field (e.g., near a bar magnet). This ubiquitous misconception was followed by misconceptions involving "Metal" (13 misconceptions), such as "every metal is magnetic", ignoring that some but not all metals are only magnetizable. Furthermore, "Iron" was involved in multiple misconceptions due to the aforementioned reasons (11 misconceptions). At timepoint two, a similar picture arose as "Metal" was involved in 15 misconceptions, followed by "Magnetic" with 14 misconceptions, and "Iron" with 12 misconceptions. At timepoint three, "Magnetic" again was involved in many misconceptions (28 misconceptions). Moreover, the "Geomagnetic Field" (25 misconceptions) and the "Picture_Gravitation" (18 misconceptions) were often connected with misconceptions. These were often labelled as being the same thing, or students mixed some characteristics between these fields.

Across all timepoints, the terms that were involved in most of the wrong connections and thus appear to be hard for the students to correctly integrate into their network were "Magnetic" (41.5% of edges), "Geomagnetic Field" (32.4% of edges), and "Cobalt" (29.3% of edges). Interestingly, also the pictorial nodes were often related to misconceptions. For example, misconceptions were that the field lines around cables represent an electric field or that the field lines for a single charge show attraction.

Regarding the central terms at timepoint three mentioned above, their involvement in misconceptions was rather low. At timepoint three, "Attraction", "Field", and "Magnetism" were involved in four misconceptions, and "Magnetic Field" as well as "Magnet" in three cases with misconceptions. Relative to the overall usage (i.e., degree) of these five concepts, this resulted in an amount of 3.5%–5.4% of misconceptions with these terms.

As the misconceptions always involved the relation of two nodes, the examination of separate terms can also be enriched by an investigation of common pairs of nodes with a misconception. The most common pairs of misconceptions across all timepoints were "Geomagnetic Field" with "Picture_Gravitation" and "Gravitational Field", since many students did not differentiate correctly between those two types of fields, and often mixed them or treated them equally. Moreover, frequent misconceptions concerned the pairs of "Magnetic" with "Nickel", "Cobalt", "Iron", and "Metal", as these materials are not magnetic by themselves, but only magnetizable when a magnet is approached. In contrast, magnets are principally magnetic and not magnetizable, which concerned another pair of misconceptions ("Magnetizable" to "Magnet"). Another common link that had multiple misconceptions was "Electric Current" and "Electric Field", as electric current does generate magnetic fields and only charges generate electric fields.

Besides misconceptions, we were interested in some edges between special terms and how these emerged across time in the idea of temporal development. The occurrence of these links is depicted in Table 3. Since one of the learning goals was to identify repulsion and attraction from field line depictions, we investigated whether students used the depictions appropriately—i.e., connecting them to attraction and repulsion respectively. In the case of magnets, the students were already able to recognize attraction and repulsion correctly at timepoint one, by considering either the orientation of the magnets or the structure of the field lines. We assume that this can be explained by the fact that all students were taught on magnetism during primary school.

The depictions of cables were connected less often to attraction and repulsion than depictions of charges, but nevertheless these were correctly connected by a majority of students at timepoint three, indicating that students were able to read and interpret field line depictions after the unit.

Another learning goal was the recognition of the common abstract structure of the three different types of force fields. Specifically, we wanted the students to know what generates each force field. To facilitate this process of abstraction, we contrasted the three fields in the last lesson and let the students work out commonalities and differences between these fields. We therefore wanted to know whether students grasped the concept that charges generate electric fields, electric current generates

magnetic fields, and mass generates gravitational fields. Except for the gravitational field, the students rather had problems in identifying electric current as the cause for magnetic fields or charges as the causes for electric fields. Moreover, we were also interested in a common connection from electric current to charges. Drawing this connection is not wrong (electric current is expressed as the flow rate of electric charge) but was not of focus in this teaching unit. At each timepoint, six concept maps contained this connection, showing that this connection from electrical physics dominates the relation which one would draw from the perspective of electrostatics, where electric current connects to magnetic fields.

**Table 3.** Specific edges between terms across the three timepoints.

|  | Timepoint 1 | Timepoint 2 | Timepoint 3 | Total |
|---|---|---|---|---|
| Picture_Magnets_Attraction—Attraction | 15 | 15 | 13 | 43 |
| Picture_Cable_Attraction—Attraction | 1 | 2 | 6 | 9 |
| Picture_Charges_Attraction—Attraction | 3 | 3 | 10 | 16 |
| Picture_Magnets_Repulsion—Repulsion | 20 | 20 | 17 | 57 |
| Picture_Cable_Repulsion—Repulsion | 2 | 2 | 7 | 11 |
| Picture_Charges_Repulsion—Repulsion | 5 | 5 | 14 | 24 |
| Charge—Electric Field | 6 | 5 | 5 | 16 |
| Electric Current—Magnetic Field | - | - | 1 | 1 |
| Mass—Gravitational Field | 2 | 2 | 14 | 18 |
| Electric Current—Charge | 6 | 6 | 6 | 18 |

### 3.2. Aggregated Concept Maps

We aggregated all 87 concept maps across the timepoints, as well as for every timepoint separately, with weighting the edges that occurred in more than one concept map. The aggregated map across all timepoints resulted in a weighted graph with 37 nodes and 384 weighted edges. We used pruning to interpret the most prominent edges and concepts and deleted all edges with a weight smaller than nine. This value was determined empirically, as the distribution of weights descriptively showed a cut-off value at this weight, as shown in Appendix A Figure A5. This resulted in pruning 81.5% of the edges. The resulting graph is shown in Figure 3, revealing a structure that was common in most of the student maps, as shown in Appendix A Figures A1–A4 (for the unpruned maps). We used different colors for the seven categories that we classified the nodes into.

Prominent edges seem to exist between "Attraction" and its respective depiction of attracting magnets as well as between "Repulsion" and its depiction of magnets. Another prominent edge is shown between "Gravitational Field" and its depiction but also to the "Geomagnetic Field", which might reveal a misconception or lack of clarity in the concepts. "Magnet" seems to be the most prominent node, which nicely reflects findings on the landmarks of the unique concept maps above. Moreover, "Attraction", "Magnetic Field", "Field Lines", "Field", "Magnetism", and "Electromagnetism" appear to be central terms. Regarding the different categories, it appears that Depictions (green) and Objects (light blue) are less central, whereas the Models of the field (orange), the Attributes (dark blue), and Topics (red) are well integrated in the concept map. This is also indicated by the mean of the betweenness centrality by these categories across time. For depictions ($B = 0.04$) and objects ($B = 0.02$) the betweenness was low, whereas models had a little higher betweenness centrality ($B = 0.07$), followed by attributes ($B = 0.1$) and topics ($B = 0.12$).

Moreover, we pruned the aggregated concept maps at each timepoint separately, to reveal a change in structure over time. The resulting graphs are shown in Figure 4. As a cut-off we used a weight of five for all timepoints, as shown in Appendix A Figure A6 (for the weight histograms), resulting in a pruning of 84.5% of the edges at timepoint one, 86.8% of the edges at timepoint two, and 82.5% of the edges at timepoint three. Across time, more and more terms got integrated into the map.

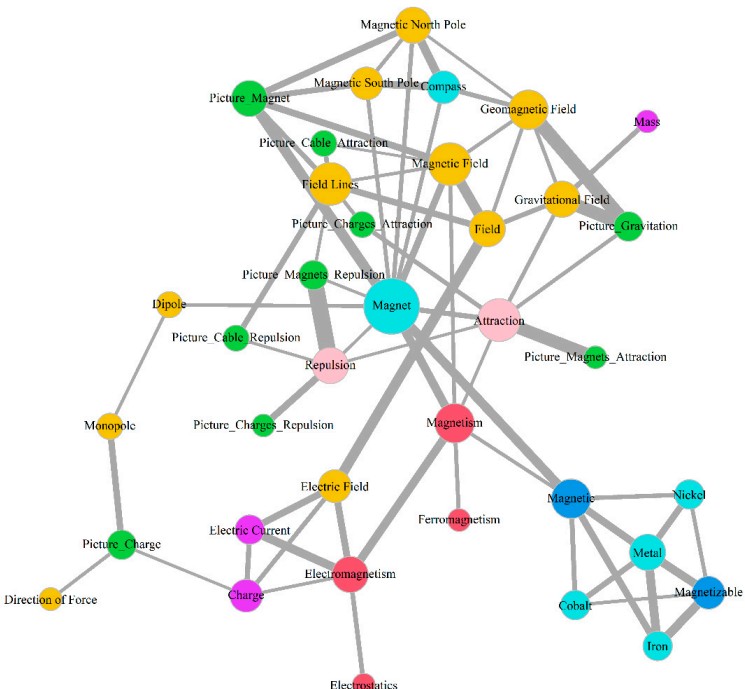

**Figure 3.** Aggregated concept map across all timepoints after pruning. Edge width is proportional to the number of occurrences in individual maps. The size of nodes is proportional to their degree. Color depicts the type of term: light blue = Objects, orange = Models, green = Depictions, purple = Physical Quantities, dark blue = Attributes, pink = Phenomena, and red = Topics.

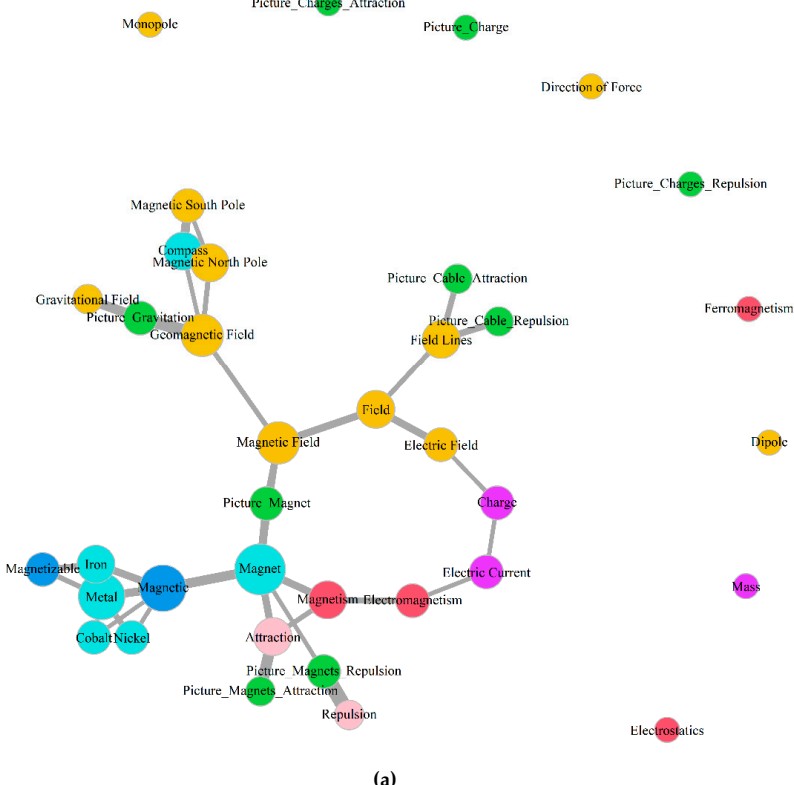

**(a)**

**Figure 4.** *Cont.*

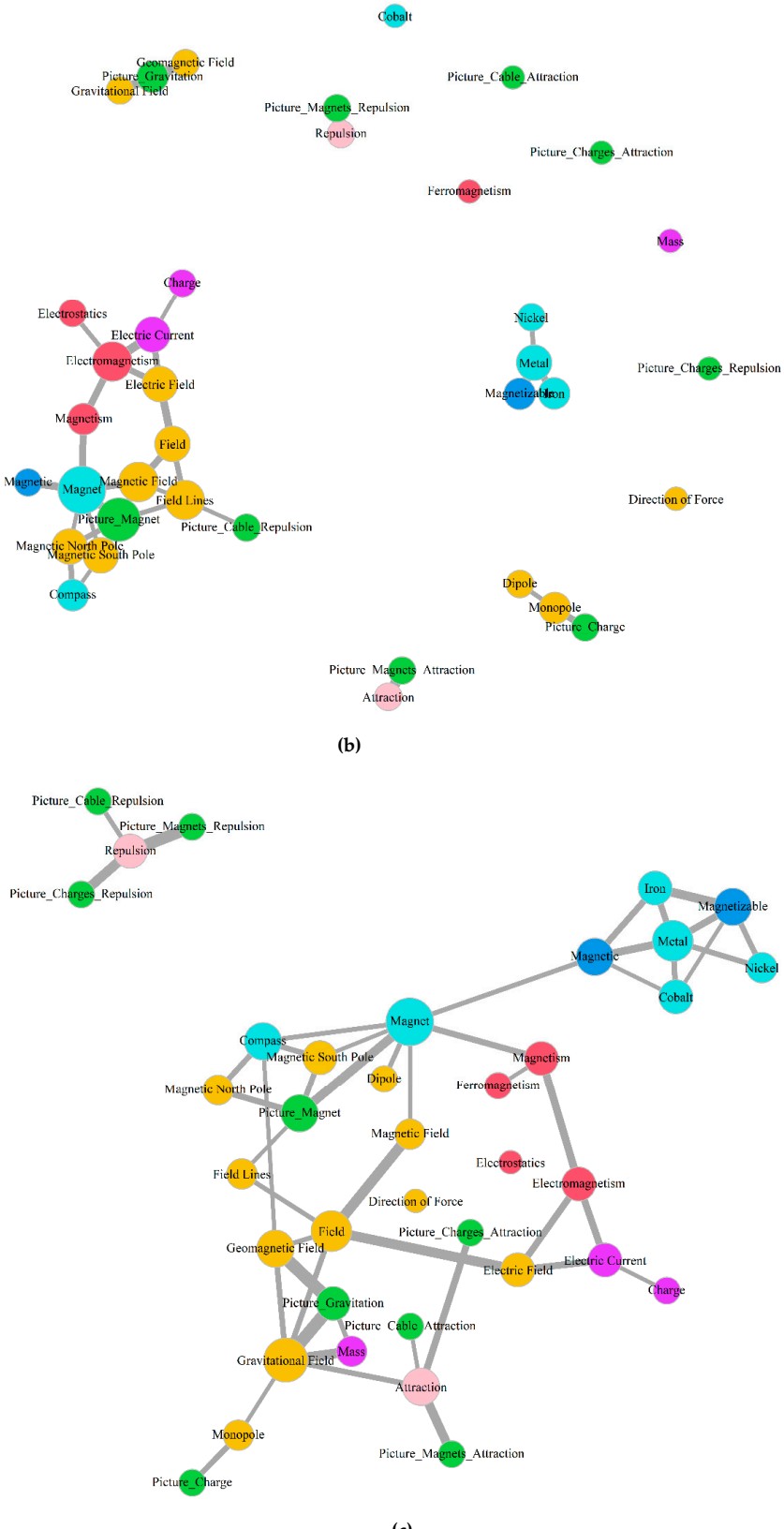

**Figure 4.** Aggregated maps per timepoint after pruning. (**a**) Aggregated map at timepoint one; (**b**) aggregated map at timepoint two; (**c**) aggregated map at timepoint three. Edge width is proportional to the number of occurrences in individual maps. The size of nodes is proportional to their degree. Color depicts the type of term: light blue = Objects, orange = Models, green = Depictions, purple = Physical Quantities, dark blue = Attributes, pink = Phenomena, and red = Topics.

As these aggregated maps reveal the structure that is common across many students, we could interpret this as a typical map at each timepoint, albeit with a little higher number of edges than from a typical student. At timepoint one, the pruned concept map showed one large component and nine items that are less incorporated. Two of the main topics, "Ferromagnetism" and "Electrostatics", remained less integrated, as well as the depictions of charges, "Dipole" and "Monopole", which were new terms for the children. Timepoint two revealed four to five small and fragmented groups of nodes, on the topics of the gravitational field, repulsion, attraction, charges, and material knowledge. That is, the single large component from timepoint one split up into more fragmented, smaller components besides one larger component that connected terms of electric and magnetic fields. At timepoint three, a single large component emerged again, integrating most of the concepts. Interestingly, it incorporated "Attraction", which is a characteristic of all three field types (electric, magnetics, and gravitational fields), but not the component of "Repulsion", as repulsion is not possible in all fields (gravitational fields show attraction but not repulsion, because there is no negative mass). Regarding the categorization of terms, one can again observe that Depictions (green) remained less connected compared to the very integrated terms of Models (orange).

### 3.3. Clustering Structure

A very peculiar aspect of concept maps is that they reveal content-specific knowledge structures. Different from simple concept tests, a network of links and edges can reveal homogeneous subgroups of highly interconnected knowledge structures. To identify such major subgroups, we applied a clustering algorithm on all student maps. For illustrative purposes, we show in Figure 5 which clusters have been identified by the clustering algorithm in the aggregated maps after pruning. Not considering the single nodes that were put in separate clusters, one can also see a tendency of an increase of clusters (from five to eight clusters to timepoint two) and then a decrease of clusters to timepoint three (six clusters). However, these figures simply serve demonstrative purposes. The analyses presented next are based on the individual maps.

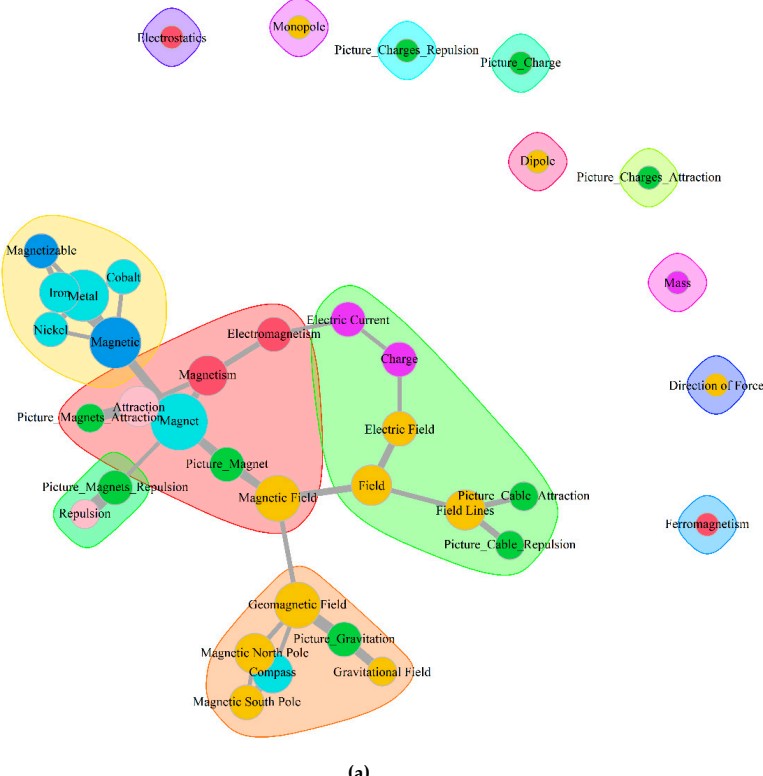

**(a)**

**Figure 5.** *Cont.*

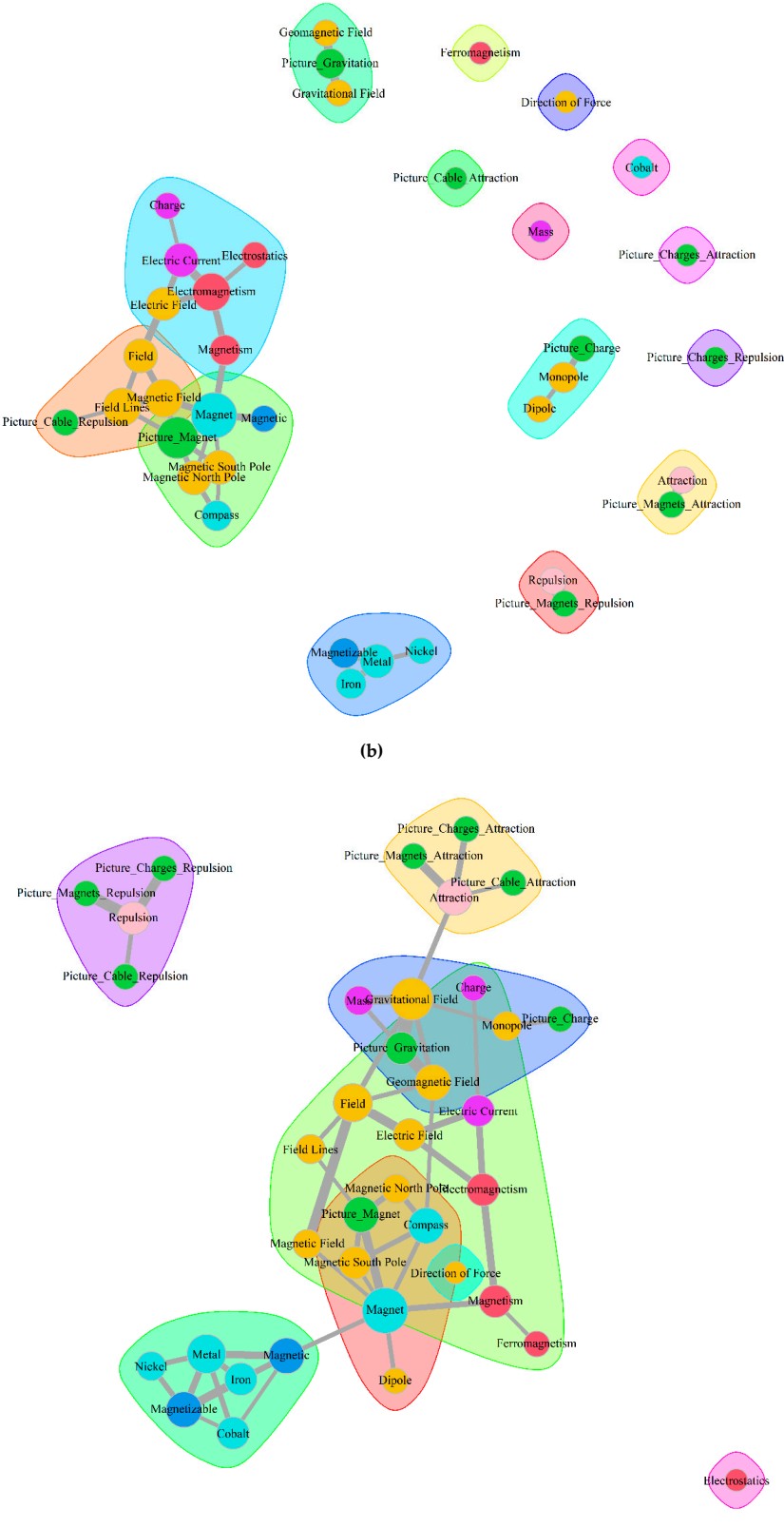

**(b)**

**(c)**

**Figure 5.** Clustering of the aggregated maps after pruning. (**a**) Clusters in aggregated map at timepoint one; (**b**) clusters in aggregated map at timepoint two; (**c**) clusters in aggregated map at timepoint three. Edge width is proportional to the number of occurrences in individual maps. The size of nodes is proportional to their degree. Color of the group depicts a cluster. Color of the node depicts the type of term: light blue = Objects, orange = Models, green = Depictions, purple = Physical Quantities, dark blue = Attributes, pink = Phenomena, and red = Topics.

Across time, the number of clusters differed significantly ($t_1 = 5.67$, $t_2 = 6.72$, $t_3 = 6.18$, $F(1.81, 48.88)$ $= 4.67$, $p = 0.02$, $\eta_G^2 = 0.05$). Figure 6 depicts the number of clusters for each individual concept map as well as boxplots for summarizing statistics. Pairwise comparisons with Tukey-adjustments for multiple comparisons revealed a significant difference from timepoint one to two ($t(54) = -3.05$, $p = 0.01$), but none from timepoint two to three ($t(54) = -1.36$, $p = 0.37$) or from timepoint one to three ($t(54) = 1.69$, $p = 0.22$). It is most likely that the non-significant pairwise comparisons resulted due to the rather small sample size and the related small power. Descriptively, the number of clusters increased from timepoint one to two, and decreased again at timepoint three, indicating a process of fragmentation followed by an integration of knowledge. The aggregated maps show such a tendency as well.

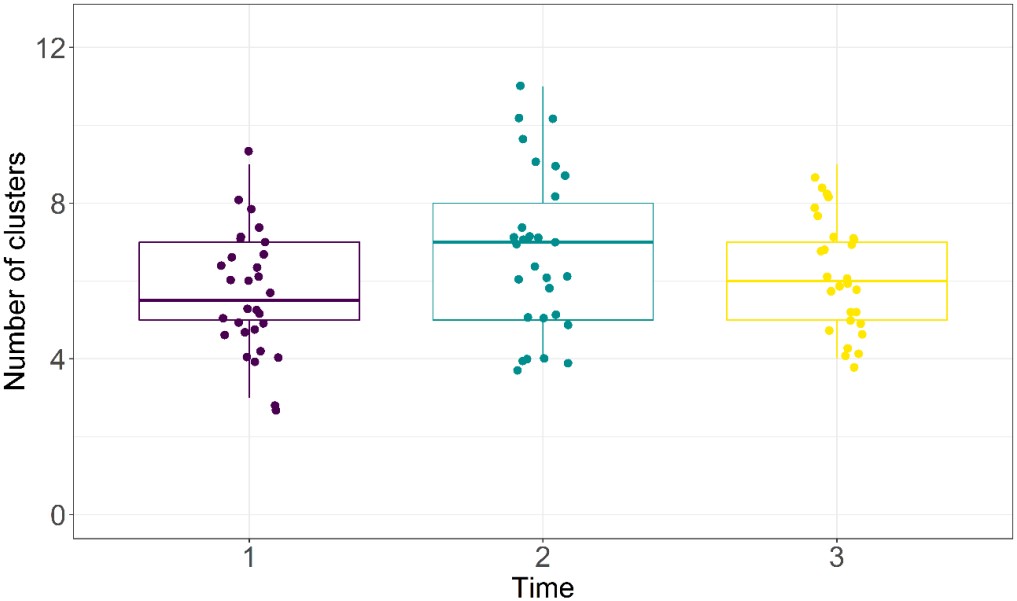

**Figure 6.** Number of clusters per timepoint.

### 3.4. Different Node Types

Already in the aggregated concept maps we investigated the manner of different node types. The visual inspection as well as the mean betweenness centrality of the aggregated maps indicated that Topics were rather central, whereas Depictions and Objects were not that strongly connected to the rest of the network. We also investigated the betweenness centrality of these node types at each timepoint on the aggregated level; Figure 7 depicts the betweenness centrality across the three timepoints. Throughout all timepoints, the terms classified as **Topics**, that is, the terms that were considered on a higher hierarchy level, had a very high betweenness centrality ($t_1 = 0.11$, $t_2 = 0.14$, $t_3 = 0.10$). Only at timepoint one, nodes that termed **Attributes** excelled the centrality of topics, to drop to timepoint two, and rise again to timepoint three ($t_1 = 0.13$, $t_2 = 0.06$, $t_3 = 0.09$), indicating that knowledge about the magnets played a central role for the students. From a physics perspective, it is rather uncommon that such attributes are central, but the structure of the clustered maps revealed that these terms formed a highly coherent separate component together with the Object nodes ("Metal", "Iron", "Cobalt", "Nickel"). This component was then connected by the attribute terms and by the term "Magnetic" to the rest of the network.

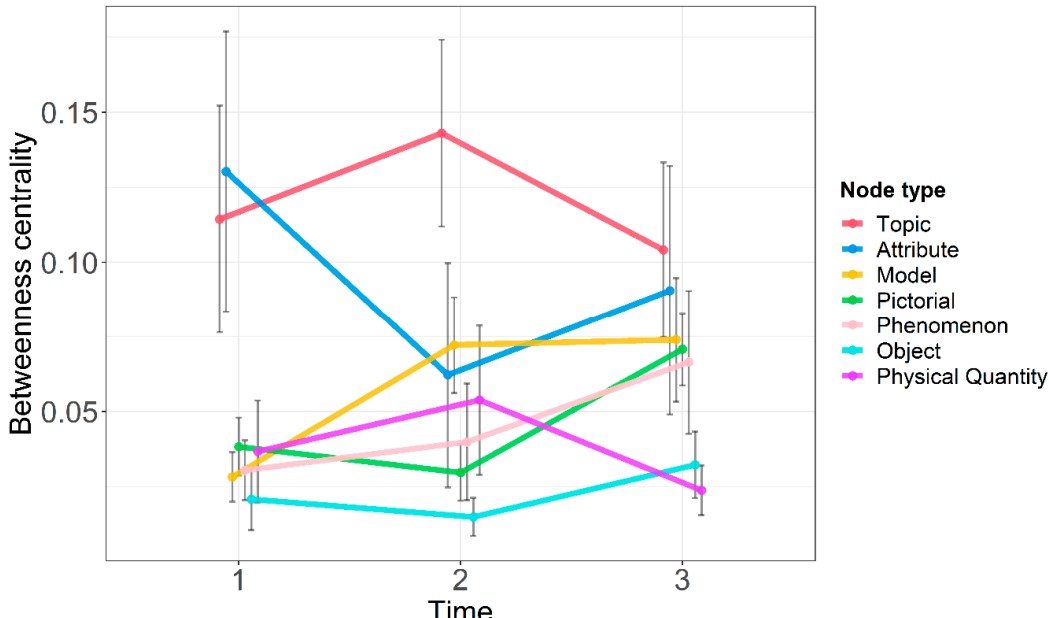

**Figure 7.** Betweenness centrality for the different node types across timepoints. Bars indicate the 95%-confidence intervals.

Those terms, which related to physical **Models** such as the field, started with a rather low centrality to become more central at timepoint two and three ($t_1 = 0.03$, $t_2 = 0.07$, $t_3 = 0.07$). **Depictions** dropped in centrality to timepoint two, to increase again at timepoint three ($t_1 = 0.04$, $t_2 = 0.03$, $t_3 = 0.07$). The centrality of **Phenomena** (repulsion and attraction) increased throughout all timepoints ($t_1 = 0.03$, $t_2 = 0.04$, $t_3 = 0.07$), indicating a higher integration into the concept maps, as the topic of the field and the interpretation of field lines depicting these phenomena was learned. **Physical Quantities** ($t_1 = 0.04$, $t_2 = 0.05$, $t_3 = 0.02$) decreased in terms of centrality, and were least central at timepoint three. The betweenness centrality of **Object** terms, that is, graspable, real world objects, stayed low, leaving them the least integrated terms at timepoint one and two ($t_1 = 0.02$, $t_2 = 0.01$, $t_3 = 0.03$). This indicates that the students had a stronger focus on more abstract terms, such as the Topic terms mentioned above.

*3.5. Comparison with Experts*

We compared the concept maps of the four experts with the student maps from all timepoints. Figure 8 shows the descriptive differences between those maps. Experts showed a higher number of edges than most of the students at all timepoints. Interestingly, they showed a smaller density, which would indicate a less connected map. At the same time, the expert maps showed a higher diameter and distance than most of the student maps. The number of clusters more or less equals the number of clusters from the students at their last timepoint.

We compared the differences between student maps at timepoint three and the expert maps using Mann–Whitney tests. There were significant differences regarding the number of edges ($U(28, 4) = 12.5$, $p = 0.01$) as well as the density ($U(28,4) = 112$, $p < 0.001$). The diameter was not significantly different between the student maps at timepoint three and the expert maps ($U(28,4) = 28$, $p = 0.11$). Moreover, the difference between the mean distance of student maps at timepoint three and expert maps was not significant ($U(28,4) = 22$, $p = 0.05$) nor was the number of clusters ($U(28,4) = 60$, $p = 0.84$).

Regarding the most central terms in the four expert maps, we again extracted the five terms with the highest betweenness centrality. Similar to the student maps, "Magnet" was the most central term. As for timepoint three for the student maps, also "Field" and "Attraction" were central. In the expert maps, "Electric Field" and "Ferromagnetism" played a central role, which was not so much the case in the student maps. The comparison of the five most central terms between expert maps and student

maps at timepoint three is shown in Table 4. Interestingly, the degree and betweenness centralities were higher than in the student maps. This follows from the fact that the expert maps had a rather high diameter and low density. Most of the terms were rather poorly integrated into the map, and only the very central terms were strongly connected. Thus, the expert maps were clearer with respect to which nodes were central and more hierarchically structured. Table A2 in Appendix A compares the centrality indices for all concepts from the student maps across the three timepoints with the centrality indices of the expert maps.

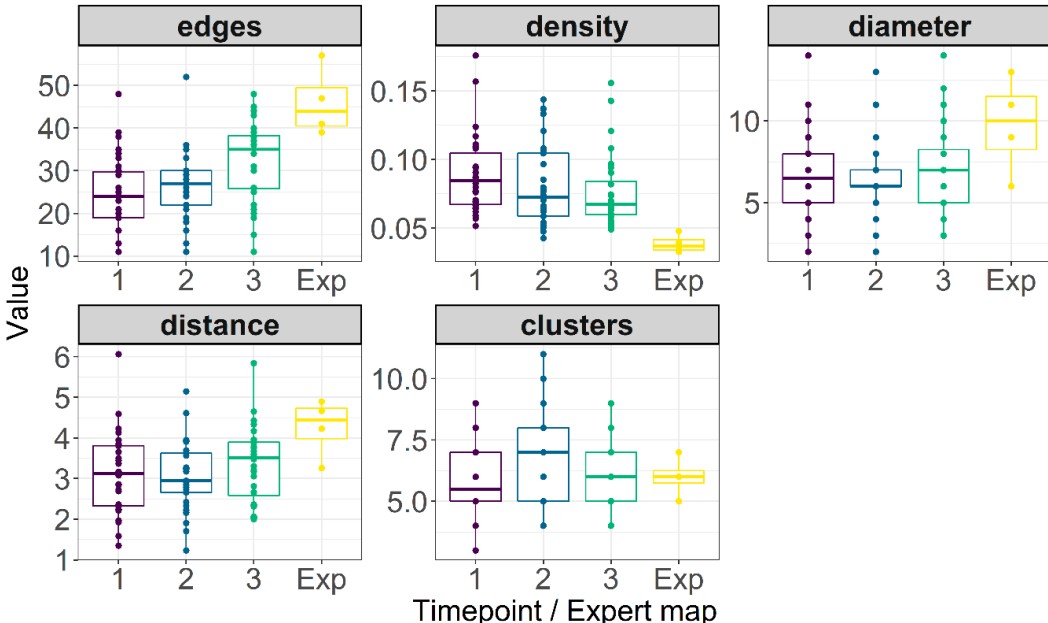

**Figure 8.** Number of edges, density, diameter, distance, and number of clusters from student maps at all timepoints compared to expert maps.

**Table 4.** Most central nodes in the student maps at timepoint three and in the expert maps indicated by betweenness centrality (B). Degree (D) and PageRank (PR) given for comparison.

| | Students (Timepoint 3) | | | | Experts | | |
|---|---|---|---|---|---|---|---|
| B | D | PR | Node | B | D | PR | Node |
| 0.2 | 3.4 | 0.042 | Field | 0.29 | 6.75 | 0.024 | Magnet |
| 0.19 | 3.74 | 0.053 | Magnet | 0.24 | 5.00 | 0.036 | Electric Field |
| 0.16 | 2.92 | 0.040 | Magnetic Field | 0.23 | 4.00 | 0.031 | Ferromagnetism |
| 0.15 | 3.21 | 0.043 | Magnetism | 0.18 | 3.75 | 0.027 | Field |
| 0.12 | 2.85 | 0.042 | Attraction | 0.17 | 4.00 | 0.058 | Attraction |

## 4. Discussion

Our study showed that while acquiring a new topic, students' concept maps differ in their structure and content across time. Using various methods of network analysis, we could investigate which methods were sensitive to such changes even in a relatively small sample of 30 students. For example, by using aggregated maps and by pruning them, we could unravel the underlying structure of the concept maps that was common for the majority of students. Furthermore, we could observe the process of fragmentation and integration across time using a clustering algorithm.

The term conceptual change is often associated with changes in the deeper, underlying knowledge. The process of conceptual change, however, comes in degrees. At the most basic level, conceptual change is associated with assimilation of new knowledge and facts into the existing knowledge structure, enriching it [6,13]. The descriptive measures of the concept maps indeed indicated that

learning progress was associated with larger networks with more edges and increased diameter. In addition, the number of unknown terms decreased across time. Taken together, these rather simple measures revealed the assimilation type of learning with new associations forming, as well as the integration of new knowledge into the existing knowledge. This was also reflected in the comparison of the students' and experts' maps, which showed that expertise was associated with larger networks (larger diameter and more edges). However, it is widely acknowledged that learning concepts goes beyond mere accretion of new knowledge and includes changes in the structure and content of knowledge. Analysis of the number of clusters and the pruned aggregated maps revealed processes of fragmentation followed by an integration of concepts. This process is only partly in line with the common view described in the literature, where conceptual change is often perceived as a rather one-way process towards a higher integration of knowledge [15]. Fragmentation and inconsistencies may happen during the course of learning when new information is added to the knowledge structure and it may take time before concepts are fully incorporated as part of the knowledge [13].

In addition to the global measures of structure, we analyzed the maps on the level of individual nodes as well. We showed how different types of nodes changed in their centrality across time. Our findings are in line with other studies examining students' concept maps, where globally important nodes are found to be the abstract ones; for example, in the context of electromagnetism the field concepts often come up as the most central [63,64]. Such concepts are applicable in a wide variety of situations, which may explain their global role in the maps. Concerning pictorial nodes, we could show that students do use them in their concept maps and can integrate such different types of nodes that convey more complex information. These depictions acted rather as support for other, more central concepts, and remained themselves less central. Nevertheless, students integrated the depictions correctly into the maps.

Several proposed schemes for coding concept maps suggest weighting different types of hierarchies differently [36,37]. For example, an abstract concept, which leads to a branch of more concrete concepts would be rated as more important than the more concrete concepts of a lower hierarchy level. Our comparison of different types of terms showed that this is a reasonable approach, as more abstract terms (i.e., which had a higher knowledge hierarchy) showed to be more central and thus important for the structure of the map.

In addition to investigating individual nodes independently, investigating how students connected different concepts allowed us to detect possible misconceptions, and we could unravel the most important misconceptions in this group of students. Misconceptions were mainly about the nature of the geomagnetic field compared to the gravitational field, as well as about the differentiation between the terms magnetic and magnetizable. Both of these misconceptions are rather specific to this study. As the teaching unit did not explicitly focus on the nature of the geomagnetic field, but only the gravitational field, students could not sufficiently contrast these two fields from each other but mixed them. Differentiating between the terms magnetic and magnetizable might have been difficult for the students, as it is a linguistic subtlety of the German language. Whereas in English it is more common to use the notions hard and soft magnetic, in German everyday language, one often speaks about an object being magnetic, as it gets attracted to a magnet. Thus, this misconception has to be viewed as a mild one, which does not necessarily hinder the acquisition and comprehension of magnetism and electrostatics.

Finally, comparing the student maps to expert maps also led to interesting results. By comparing the general network analysis measures, the experts showed a higher number of edges, thus drawing more connections between the topics, but at the same time a larger diameter and a lower density, indicating a structure that more clearly focuses around central concepts. The principle that novices rather tend to focus on surface features and experts on core properties [1,3] could be demonstrated in this study as well. While students at timepoint one focused on attributes and terms related to material knowledge of magnets and gave them very central positions within their concept maps, this changed to timepoint three, where models such as the field became more central, similarly to the expert maps.

### 4.1. Usage of Concept Mapping in Magnetism and Electrostatics in School

As mentioned above, concept maps can be used as a diagnostic instrument for teachers as well as a learning tool for students at the same time [10,65,66]. Regarding the learning process, the construction of a concept map initiates the formation of an overall perspective onto the topic by rendering complex relations of notions into a structured graphical representation. It allows students to summarize what they already know about a topic and to integrate recently learned concepts intentionally into their existing knowledge base by interrelating different notions [32].

For teachers, the ability to judge their students' learning progress is of utmost importance. Depending on the way the students constructed the concept map, there exist different methods to evaluate students' learning progress [67]. In this study, we used measures of graph theory to analyze the concept maps. Such network analyses are not always applicable in the school context. Thus, we exemplify a few methods which teachers can use in the classroom to evaluate concept maps that have been constructed from a fixed set of notions.

In a first step, teachers could examine what notions are unknown before instruction. To survey to what extent new notions are understood, teachers could examine whether new notions are integrated in a meaningful way into the concept map and check on possible misconceptions. Likewise, such a diagnosis can also be made by considering important terms with a high centrality if measures of centrality are applicable.

Teachers can also single out specific relations which are either typical of novices—that is, relations that express misconceptions or demonstrate the fact that knowledge is structured by surface criteria [3]. Alternatively, teachers can focus on specific relations which are typical of experts and which show that the knowledge structure is organized by abstract principles as exemplified in Table 3. Such an analysis gives information to what extent the students' knowledge structure is transformed towards an expert's one.

Concept maps can also be used to promote metacognitive strategies [68] by requiring students to compare their final concept map with an expert map. In order to facilitate the comparison, notions of specific interest can be highlighted, or clusters can be outlined. Students can be prompted to work out important differences between the two maps and reflect by which criteria the concept maps are organized. Such a comparison allows students to realize how their own perspective on the topic differs from an expert perspective and helps them to integrate new knowledge better [69].

### 4.2. Limitations

This study investigated concept maps from a highly selective sample with students who had a lot of prior knowledge in physics and especially in the topic of magnetism. Consequently, these results have to be regarded as exploratory. Moreover, the learning setting in this study differed from a typical class setting, as we taught students in small groups and outside of regular classes. Enrolling in our lessons was voluntary. Thus, the sample was probably more motivated than a usual class. We therefore assume that the results do not generalize to the whole student population. They nevertheless inform us about the knowledge structure that students have, when they already invested some effort in learning the topic. Moreover, the comparison with the expert maps showed that also those concept maps from students with high prior knowledge do differ from the concept maps that experts draw and thus these concept maps probably represent an intermediate step between the knowledge structure of students with no prior knowledge and the knowledge structure of experts. As the number of experts was small and their coherence in the maps was also rather low, it is unclear how reliable the observed differences between novices and experts are found with other samples. Thus, an avenue for future research lies in a comparison of a larger group of experts against a group of novices. We aim at conducting such a study in the future and we will test the impact of prior knowledge on this specific topic in a further project (see Section 4.3).

In this research, the integration of knowledge might have been particularly facilitated by the special nature of the teaching unit. Integrating new terms into an existing knowledge network often does not

happen spontaneously. Indeed, the methods used in this unit, such as contrasted comparisons [70], self-explanations, and different types of learning opportunities (experiments, thought experiments, direct instruction, texts) aimed directly at the integration of knowledge. Thus, in other studies, where the final integration is not so much in focus in the teaching material, the process of integration might not be observed to the extent observed here. Only future work using different topics and also different teaching methods will increase the validity of these exploratory findings.

Regarding the classification of the node types, one has to bear in mind that there was a different number of words per type and that the categorization strongly depends on the way the concept map is constructed. Slightly different terms or a slightly different categorization could have changed the centrality measures. Thus, the meaning of the allocated types is not generalizable with regard to content and learning. However, the observation that more abstract terms such as "Topics" tended to be more central, supported the reasoning of other coding schemes for concept maps, which often allocate different scores for terms at different hierarchy levels.

Another limitation concerns the stability of the knowledge restructuring that occurred. As our design did not include later follow-up testing, we cannot infer that the concept maps at timepoint three depict the final stage of the learning process. Indeed, some restructuring might happen later after the teaching unit. This study, however, showed that already in a rather short amount of time (about two weeks) and few teaching units (five lessons of 45 min), a change in the knowledge structure took place—the concept maps got more and more similar to the expert maps across time. Future research might investigate the stability of concept maps when no active teaching/learning is involved and use such a unit on a broader range of students to test the generalizability.

*4.3. Outlook*

As this study is embedded in a larger project (https://osf.io/xkem5/), we will test the exploratory results from this study again in a confirmatory manner after the data collection for the whole project has finished. We will test whether the process of fragmentation and integration can be replicated with a larger sample, including students with high and low prior knowledge on the topic. We will then also investigate the role of prior knowledge on the process of conceptual change in this topic.

**Author Contributions:** Conceptualization, C.M.T., B.H. and T.K.; data curation, C.M.T.; formal analysis, C.M.T.; investigation, C.M.T.; methodology, C.M.T. and B.H.; project administration, C.M.T.; software, C.M.T.; validation, C.M.T.; visualization, C.M.T.; writing—original draft, C.M.T. and T.K.; writing—review & editing, C.M.T., B.H. and T.K. All authors have read and agreed to the published version of the manuscript.

**Funding:** Tommi Kokkonen is funded by the Academy of Finland, grant number 311449. The APC was funded by the ETH library. Other than that, this research received no external funding.

**Acknowledgments:** We thank Elsbeth Stern (E.S.) for supporting this project with resources, supervision and excellent tips, Stefan Wehrli (S.W.) for organizational and technical support and Tobias Beckert for piloting the lecture material, for commenting on the lessons as well as on the concept mapping task.

**Conflicts of Interest:** The authors declare no conflict of interest.

# Appendix A

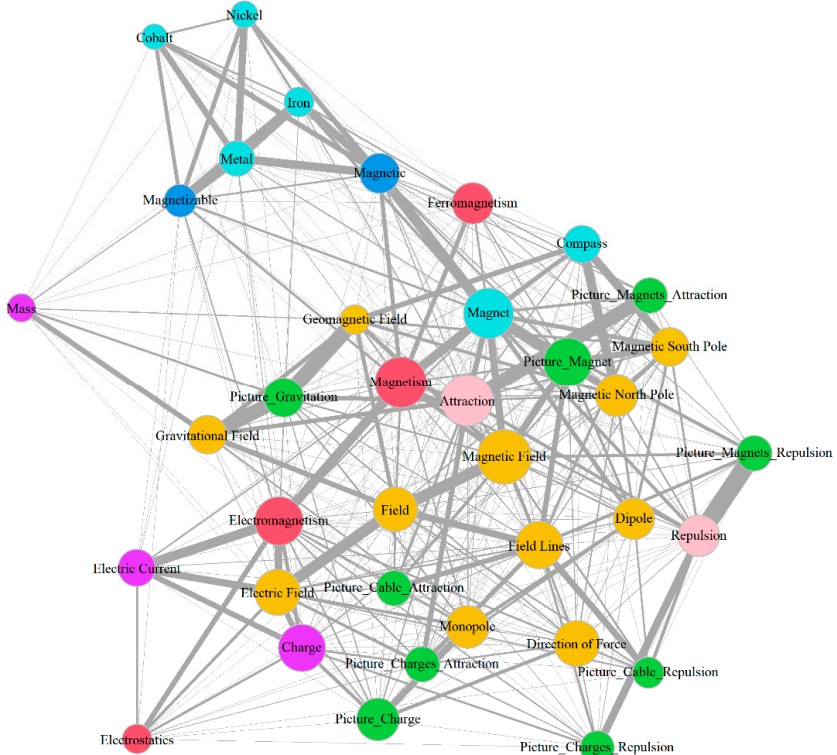

**Figure A1.** Aggregated concept map across all timepoints with no pruning. Color depicts the type of term: light blue = Objects, orange = Models, green = Depictions, purple = Physical Quantities, dark blue = Attributes, pink = Phenomena, and red = Topics.

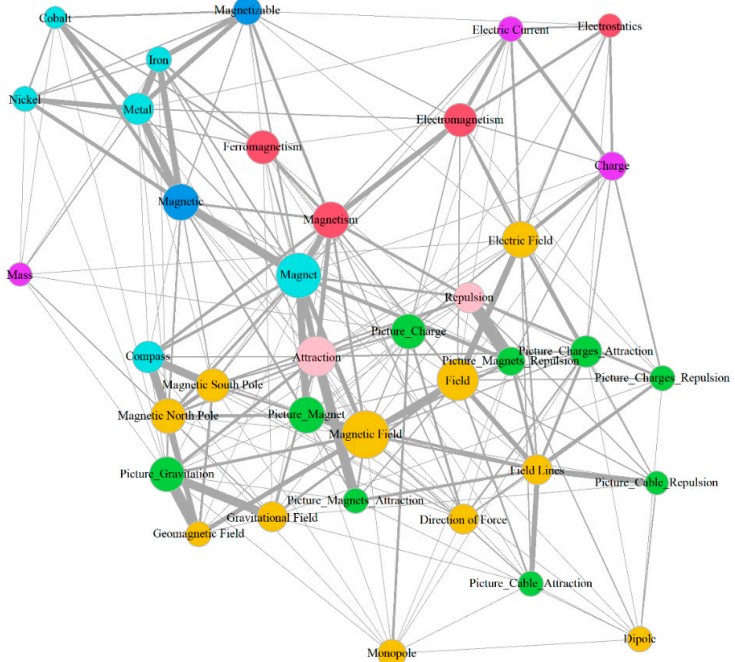

**Figure A2.** Aggregated concept map at timepoint one with no pruning. Color depicts the type of term: light blue = Objects, orange = Models, green = Depictions, purple = Physical Quantities, dark blue = Attributes, pink = Phenomena, and red = Topics.

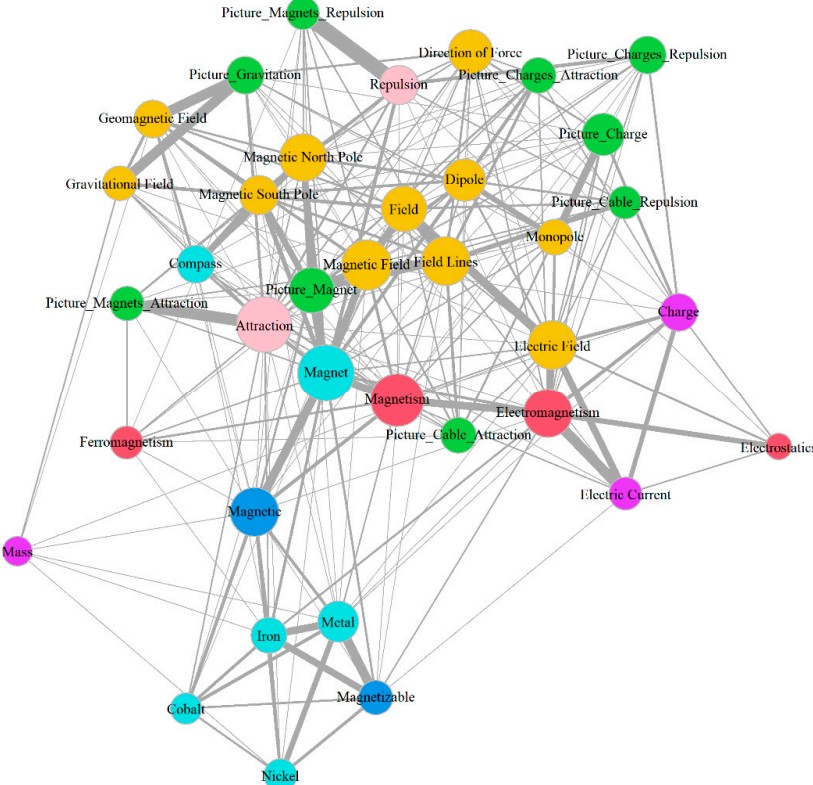

**Figure A3.** Aggregated concept map at timepoint two with no pruning. Color depicts the type of term: light blue = Objects, orange = Models, green = Depictions, purple = Physical Quantities, dark blue = Attributes, pink = Phenomena, and red = Topics.

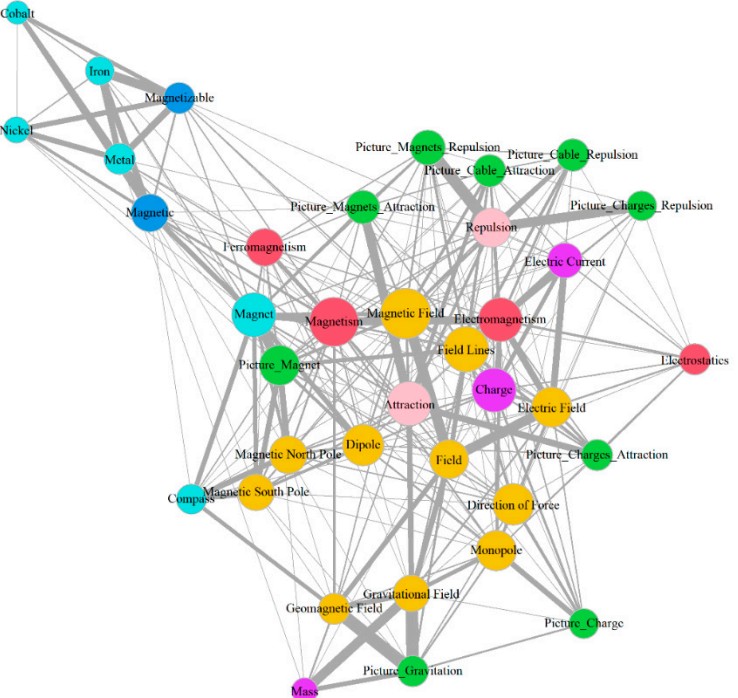

**Figure A4.** Aggregated concept map at timepoint three with no pruning. Color depicts the type of term: light blue = Objects, orange = Models, green = Depictions, purple = Physical Quantities, dark blue = Attributes, pink = Phenomena, and red = Topics.

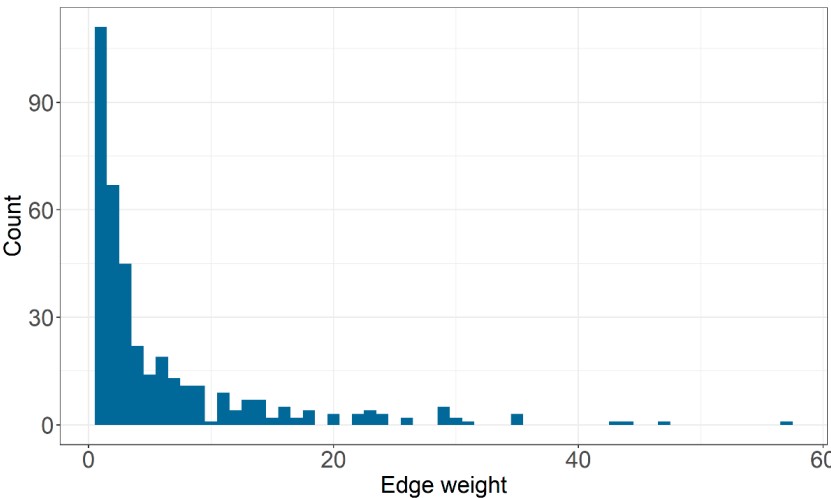

**Figure A5.** Weight distribution of links in aggregated concept map across all timepoints. A cut-off at a weight of 9 is visible.

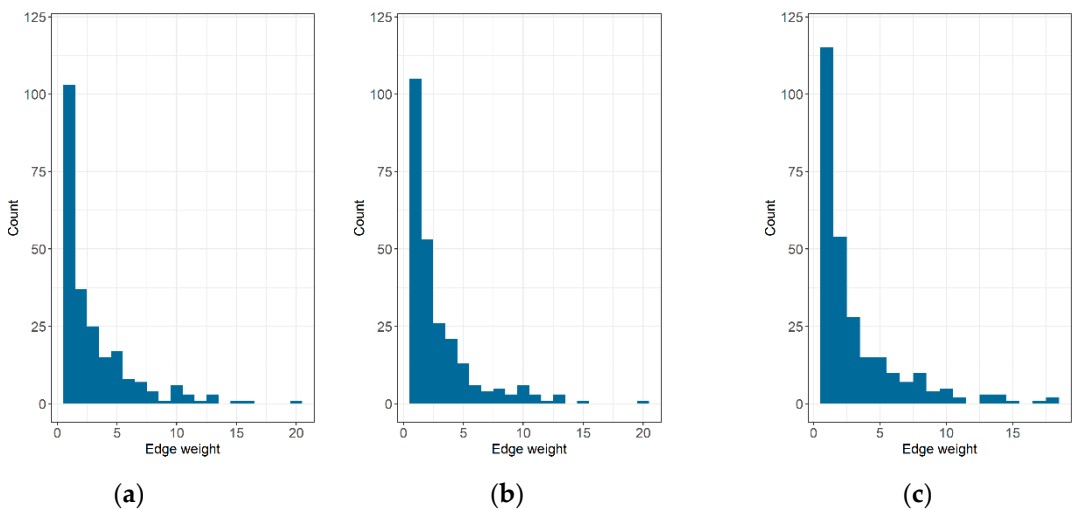

**Figure A6.** Weight distribution of links in the concept maps at each timepoint (**a**) = timepoint one, (**b**) = timepoint two, (**c**) = timepoint three. A cut-off at a weight of 5 was chosen for all timepoints.

**Table A1.** Descriptive statistics for general graph measures across timepoints. Standard deviation in parentheses. $CI_{95}$ depicts the 95%-confidence interval.

|  | **Timepoint 1** | **Timepoint 2** | **Timepoint 3** |
|---|---|---|---|
| Number of concept maps | 30 | 29 | 28 |
| Number of concepts | 24.9 (6.32) $CI_{95}$[22.6; 27.3] | 27.1 (6.20) $CI_{95}$[24.8; 29.5] | 30.4 (6.62) $CI_{95}$[27.8; 32.9] |
| Number of edges | 25.1 (8.58) $CI_{95}$[21.9; 28.3] | 26.9 (8.21) $CI_{95}$[23.8; 30.1] | 32.5 (9.61) $CI_{95}$[28.7; 36.2] |
| Density | 0.089 (0.0286) $CI_{95}$[0.078; 0.01] | 0.082 (0.0301) $CI_{95}$[0.07; 0.093] | 0.077 (0.0267) $CI_{95}$[0.067; 0.087] |
| Mean distance | 3.13 (0.994) $CI_{95}$[2.76; 3.51] | 3.05 (0.849) $CI_{95}$[2.73; 3.37] | 3.36 (0.927) $CI_{95}$[3.00; 3.72] |
| Diameter | 6.63 (2.47) $CI_{95}$[5.71; 7.56] | 6.41 (2.16) $CI_{95}$[5.59; 7.24] | 7.25 (2.61) $CI_{95}$[6.24; 8.26] |
| Number of clusters | 5.67 (1.47) $CI_{95}$[5.12; 6.22] | 6.72 (2.07) $CI_{95}$[5.94; 7.51] | 6.18 (1.44) $CI_{95}$[5.62; 6.74] |

**Table A2.** Centrality of all nodes at the three timepoints and in the expert maps ordered by betweenness centrality (B). Degree (D) and PageRank (PR) given for comparison.

| Timepoint 1 | | | | Timepoint 2 | | | | Timepoint 3 | | | | Expert Maps | | | |
|---|---|---|---|---|---|---|---|---|---|---|---|---|---|---|---|
| B | D | PR | node | B | D | PR | node | B | D | PR | node | B | D | PR | node |
| 0.265 | 3.14 | 0.058 | Magnetism | 0.183 | 2.76 | 0.044 | Magnetism | 0.196 | 3.4 | 0.042 | Field | 0.289 | 6.75 | 0.024 | Magnet |
| 0.192 | 3.32 | 0.061 | Magnet | 0.166 | 3 | 0.049 | Ferromagnetism | 0.187 | 3.74 | 0.053 | Magnet | 0.24 | 5 | 0.037 | Electric Field |
| 0.171 | 2.71 | 0.053 | Field | 0.164 | 3.7 | 0.061 | Magnet | 0.158 | 2.92 | 0.04 | Magnetic Field | 0.234 | 4 | 0.031 | Ferromagnetism |
| 0.151 | 2.54 | 0.048 | Magnetic Field | 0.143 | 3.29 | 0.053 | Electromagnetism | 0.154 | 3.21 | 0.043 | Magnetism | 0.182 | 3.75 | 0.027 | Field |
| 0.13 | 2.79 | 0.054 | Magnetic | 0.119 | 2.38 | 0.042 | Magnetic Field | 0.12 | 2.85 | 0.042 | Attraction | 0.166 | 4 | 0.058 | Attraction |
| 0.127 | 2.33 | 0.044 | Picture_Magnet | 0.1 | 2.52 | 0.044 | Electric Field | 0.108 | 2.85 | 0.042 | Gravitational Field | 0.153 | 2.75 | 0.023 | Magnetism |
| 0.116 | 3.05 | 0.056 | Field Lines | 0.098 | 2.5 | 0.046 | Field | 0.104 | 3.13 | 0.043 | Electromagnetism | 0.152 | 2.25 | 0.013 | Compass |
| 0.114 | 2.33 | 0.041 | Electromagnetism | 0.086 | 2.38 | 0.044 | Field Lines | 0.097 | 2.67 | 0.041 | Picture_Magnet | 0.146 | 3.75 | 0.05 | Repulsion |
| 0.11 | 2.36 | 0.044 | Attraction | 0.076 | 2.36 | 0.043 | Attraction | 0.096 | 2.5 | 0.036 | Electric Field | 0.139 | 2 | 0.023 | Picture_Magnets_Attraction |
| 0.106 | 2.4 | 0.046 | Electric Field | 0.075 | 2.44 | 0.042 | Picture_Magnet | 0.091 | 2.29 | 0.042 | Magnetic | 0.137 | 2.25 | 0.019 | Dipole |
| 0.102 | 2.45 | 0.046 | Ferromagnetism | 0.072 | 2.35 | 0.039 | Dipole | 0.086 | 2.64 | 0.039 | Field Lines | 0.134 | 2.75 | 0.041 | Field Lines |
| 0.075 | 1.85 | 0.041 | Compass | 0.062 | 2.45 | 0.041 | Magnetic | 0.074 | 1.91 | 0.029 | Dipole | 0.119 | 2.25 | 0.017 | Geomagnetic Field |
| 0.056 | 2.29 | 0.047 | Metal | 0.055 | 1.72 | 0.038 | Gravitational Field | 0.071 | 2.37 | 0.039 | Picture_Gravitation | 0.119 | 2 | 0.026 | Picture_Charges_Repulsion |
| 0.051 | 1.92 | 0.039 | Magnetic North Pole | 0.054 | 1.83 | 0.034 | Charge | 0.067 | 2.62 | 0.042 | Repulsion | 0.117 | 3.25 | 0.047 | Magnetic Field |
| 0.051 | 1.75 | 0.036 | Magnetizable | 0.049 | 1.54 | 0.031 | Picture_Charges_Repulsion | 0.06 | 2.5 | 0.032 | Ferromagnetism | 0.112 | 4 | 0.033 | Magnetizable |
| 0.049 | 1.37 | 0.029 | Electric Current | 0.045 | 1.55 | 0.031 | Picture_Charge | 0.052 | 2.39 | 0.034 | Geomagnetic Field | 0.107 | 2.5 | 0.042 | Magnetic |
| 0.048 | 1.57 | 0.031 | Picture_Magnets_Attraction | 0.044 | 1.6 | 0.031 | Compass | 0.05 | 2.04 | 0.028 | Monopole | 0.098 | 1.75 | 0.021 | Picture_Magnet |
| 0.046 | 2.14 | 0.034 | Monopole | 0.04 | 1.93 | 0.038 | Repulsion | 0.049 | 2.04 | 0.037 | Magnetizable | 0.091 | 3 | 0.02 | Charge |
| 0.044 | 1.68 | 0.037 | Gravitational Field | 0.039 | 2.08 | 0.039 | Metal | 0.045 | 2.25 | 0.038 | Metal | 0.089 | 2.75 | 0.028 | Gravitational Field |
| 0.043 | 1.88 | 0.043 | Geomagnetic Field | 0.038 | 1.65 | 0.03 | Direction of Force | 0.035 | 1.79 | 0.03 | Picture_Magnets_Repulsion | 0.073 | 2.5 | 0.024 | Monopole |
| 0.039 | 1.82 | 0.035 | Electrostatics | 0.037 | 1.81 | 0.034 | Magnetizable | 0.035 | 1.56 | 0.026 | Picture_Charge | 0.067 | 2.75 | 0.025 | Electromagnetism |
| 0.038 | 1.89 | 0.042 | Picture_Gravitation | 0.035 | 1.5 | 0.03 | Picture_Cable_Repulsion | 0.034 | 2 | 0.028 | Magnetic South Pole | 0.038 | 2 | 0.018 | Electrostatics |
| 0.038 | 1.53 | 0.036 | Picture_Charges_Attraction | 0.03 | 1.85 | 0.034 | Magnetic South Pole | 0.033 | 1.63 | 0.025 | Direction of Force | 0.038 | 1.75 | 0.021 | Picture_Charges_Attraction |
| 0.037 | 1.8 | 0.04 | Picture_Magnets_Repulsion | 0.03 | 1.75 | 0.037 | Picture_Gravitation | 0.032 | 1.64 | 0.028 | Iron | 0.031 | 1.25 | 0.044 | Direction of Force |
| 0.037 | 1.63 | 0.035 | Charge | 0.028 | 1.96 | 0.036 | Magnetic North Pole | 0.031 | 1.68 | 0.027 | Picture_Magnets_Attraction | 0.029 | 2 | 0.04 | Electric Current |
| 0.035 | 1.67 | 0.035 | Direction of Force | 0.026 | 1.68 | 0.032 | Monopole | 0.03 | 1.61 | 0.027 | Compass | 0.023 | 1.5 | 0.022 | Picture_Magnets_Repulsion |
| 0.03 | 1.62 | 0.038 | Repulsion | 0.026 | 1.68 | 0.032 | Electric Current | 0.029 | 1.6 | 0.026 | Electric Current | 0.022 | 1.75 | 0.019 | Picture_Charge |
| 0.029 | 1.75 | 0.035 | Magnetic South Pole | 0.024 | 1.41 | 0.03 | Picture_Magnets_Attraction | 0.028 | 1.92 | 0.027 | Magnetic North Pole | 0.015 | 2 | 0.011 | Cobalt |
| 0.028 | 2 | 0.031 | Dipole | 0.022 | 1.74 | 0.038 | Geomagnetic Field | 0.026 | 1.6 | 0.031 | Picture_Cable_Attraction | 0.015 | 2 | 0.011 | Iron |
| 0.021 | 1.77 | 0.034 | Picture_Charges_Repulsion | 0.022 | 1.31 | 0.029 | Picture_Cable_Attraction | 0.024 | 1.48 | 0.023 | Picture_Charges_Attraction | 0.015 | 2 | 0.011 | Nickel |
| 0.021 | 1.56 | 0.034 | Picture_Charge | 0.019 | 1.33 | 0.027 | Picture_Charges_Attraction | 0.024 | 2.18 | 0.033 | Charge | 0.012 | 1.5 | 0.016 | Mass |
| 0.021 | 1.48 | 0.032 | Iron | 0.016 | 1.28 | 0.028 | Picture_Magnets_Repulsion | 0.022 | 1.5 | 0.029 | Picture_Cable_Repulsion | 0.005 | 2.75 | 0.025 | Metal |
| 0.013 | 1.5 | 0.028 | Picture_Cable_Attraction | 0.015 | 1.88 | 0.033 | Iron | 0.02 | 1.33 | 0.02 | Electrostatics | 0.003 | 2.25 | 0.057 | Magnetic North Pole |
| 0.006 | 1.4 | 0.029 | Picture_Cable_Repulsion | 0.013 | 1.62 | 0.025 | Cobalt | 0.015 | 1.3 | 0.022 | Picture_Charges_Repulsion | 0 | 1 | 0.021 | Picture_Gravitation |
| 0.004 | 1.47 | 0.028 | Cobalt | 0.003 | 1.25 | 0.025 | Electrostatics | 0.009 | 1.26 | 0.022 | Mass | - | - | - | Magnetic South Pole |
| 0.002 | 1.35 | 0.026 | Nickel | 0.001 | 1.47 | 0.024 | Nickel | 0.001 | 1.23 | 0.023 | Nickel | - | - | - | Picture_Cable_Attraction |
| 0.001 | 1.43 | 0.023 | Mass | 0 | 1.14 | 0.021 | Mass | 0 | 1.14 | 0.021 | Cobalt | - | - | - | Picture_Cable_Repulsion |

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
