# Peer review of "Concept Mapping in Magnetism and Electrostatics: Core Concepts and Development over Time"

_education, doi:10.3390/educsci10050129_

Round 1

Reviewer 1 Report

The authors have improved the paper and have used more nuanced and precise language where it was needed. The introductory sections now read better. The purpose of the mapping intervention is also more clearly expressed. The section on concept mapping is improved, but does not acknowledge any wider perspective in the contemporary mapping literature, such as more recent works by Cañas. Though I feel this will limit the citation interest in this paper, that is the authors’ choice.

In assuming that physics is organized in a network-like structure [line 131], it may be helpful to cite the work of Janet Donald at this point.

Lines 763 – 767: Rather than just getting more data (i.e. a larger group of experts), it may be worth considering improving the quality of the data by interrogating the nature of expert knowledge structures more deeply. Experts do not always agree, and simply having more expert structures will introduce variability in the data reflecting differences in expert biographies. The language used and the academic/professional contexts of the experts will influence their knowledge structures (see for example Heron et al, 2018, Educational Research, 60(4), 373 – 389). 

I respect the authors decision to use network analysis to standardize their map interpretations, I would disagree with them about the importance of visual inspection to verify any interpretations. This data is neither context-free nor value-free, and so total reliance on the numbers is risky. Expert observation is the cornerstone of scientific work, and visual verification of data is important here.

Author Response

Dear Reviewer,

Thank you again for the time and effort that you invested in this second round of our manuscript.

We appreciate the positive feedback and helpful recommendations for further literature. We added a reference to Janet Donald to refer to the network-like structure of physics knowledge (line 148). Thank you as well for the reference to Heron et al., 2018; we will consider this in our future study with experts.

Furthermore, we clarified our view on the added value of network analysis on visual inspection, incorporating the aspects of context (lines 300-303).

All changes are highlighted by the track changes function for transparency.

Reviewer 2 Report

I think it is a good job and the procedure is rigorous.
Authors' comments to reviewer 3's considerations are timely

Author Response

Dear Reviewer,

Thank you again for your comments on our manuscript. We appreciate your positive feedback. We carefully included all comments in the revised version. All changes are highlighted by the track-changes function.

This manuscript is a resubmission of an earlier submission. The following is a list of the peer review reports and author responses from that submission.

Round 1

Reviewer 1 Report

Dear Authors,

Thank you for the opportunity to review the manuscript. I enjoyed reading it. The study was well designed and conducted. The analyses were comprehensive. The manuscript was well organized and well-written. I admire the authors’ work. However, I would like to share some of my comments on the theoretical framework, that is, conceptual change, presented in this paper.

This paper uses conceptual change as the main framework to collect data, conduct analysis, and interpret results. The study uses three time points of a curriculum on electromagnetism to collect student concept maps through a software with concepts and relationships provided. The concept maps were analyzed using network analysis and patterns of the three time points were compared. In addition, students’ concept maps were compared with experts’.

The conceptual change theories emphasize structural changes in leaners’ mental model. The typical conceptual change theories denote that the new and old mental models are fundamentally different: the old and new models are incommensurable; the changes from old to new model (i.e., conceptual change) are irreversible. In other words, the incremental, continuous changes in mental model, such as adding new information to existing structure and re-organization of mental model using existing fragments (such as p-prims), are not typically categorized as conceptual change. Most of the conceptual change literature addresses radical, discontinuous changes. Examples include Chi’s ontological shifts of a concept (e.g., work as a process vs. work as a status, the latter is the misconception) and Vosniadou’s hybrid model (kids’ misconception about earth with a round shell and a flat plate inside).

Other empirical examples in PER include students’ misconception of impetus (more force, more motion; no force, no motion), which should be replaced by the correct conception of Newtonian kinetics (force makes changes in motion). According to conceptual change theories, students’ misconceptions are robust. Conceptual change, therefore, happens through the process of elicit (misconceptions)àconfrontàreplace. (rather than accumulation of pieces of knowledge and reorganization)

There is the counterpart towards conceptual change, which emphasizes the instability of student ideas and the emergent changes in student understanding. This school/camp stands the opposite of conceptual change. The knowledge in pieces (p-prims) framework (by diSessa) and the resources framework (by Hammer) fall in this side of the argument. According to this knowledge in pieces point of view, learning occurs through re-organizing pieces (rather than replacing old structures).

In the literature review section of the present paper, p-prims is grouped in the conceptual change literature (on page 2), which is a little unusual. The authors need to provide more elaborated argument on this if they want to include the p-prim theory as one of the conceptual change theories.

In relation to that, if the paper takes the stance of conceptual change (it seems so), the reader would expect to see more emphasis on the structural difference in student understanding at different time points, or that between students’ and experts’. However, the data in this paper (presented in the result section) were unclear about what kind of structural changes were claimed “conceptual change.”  It seems the authors infer the changes in the nodes and edges were, but the argument was not explicit nor clear. To me, changes in the nodes and edges were not necessarily the changes in student conceptual understanding of the topic. They can be, but the author need to provide more explicit argument on it. Also, on page 18, in the first paragraph of discussion section, the authors claim that assimilation is one kind of conceptual change. The term assimilation in Piaget’s theory denotes a learning process not involving structural change of mental models. And conceptual change (as the authors cited in that particular sentence in reference 6 and 13) requires structural changes. This seems contradicting to itself.

In general, I like the study and I enjoyed reading the results. But the theoretical argumentation is not strong for publication.

Some minor comments:

The research questions are too many. Some of them can be grouped together.

On page 12, the last sentence of the first paragraph has a typo.

On page 16, the end of the first paragraph is a little hard to understand: what’s described there does not seem straightforward in the graph.

I hope my comments are helpful. Thank you again for the opportunity.

Sincerely yours,

Reviewer

Reviewer 2 Report

The authors address some interesting issues here and also offer some methodological innovation that may be worth exploring more. The paper sets out its stall with a good review of the literature on conceptual change, though there is little innovation offered here, no fresh perspective – drawing as it does on well know sources and key authors. This introduction is solid enough.

However, the paper begins to unravel a little later. There are too many research questions going in too many directions here. There are also some considerable methodological weaknesses in the application of concept mapping.

The research phase covers only three teaching sessions. As such this is a very brief intervention with a relatively small number of students. Any statistical inference from such a study needs to be treated with caution, and it may have been better to see this as a pilot study that could have opened up the research questions and which would have benefitted from some in-depth qualitative analysis of the students’ learning and their perceptions of their conceptual change.

The ‘training’ of the students needs much more explanation (lines 221 – 231). Completing the incomplete map of ‘Sound and Instruments’ would have shown them how to structure a map. We need to know what this looked like to see how this would have impacted on their perceptions of a ‘good’ map. What discussion was there in class about the use of the maps and the variety of possible answers? If maps were not integral to the curriculum, they may be separated from the teaching by different epistemologies - causing cognitive conflict among the students.

The authors make the same mistake as many researchers who look at concept mapping. The needs of the researchers (concept mapping for research) are not the same as the needs of the learner (concept mapping for learning).   Learning is not the same as ‘enlarging a network’ [line 125]. Sometimes learning can result in the editing and shrinking of a concept map, as students use more precise language in their maps and arrange their concepts in a more coherent structure. Was the teaching structured to support the development of a coherent map structure? How were the concepts introduced? What opportunities did the students have to articulate their understanding - or was this a passive, transmission-type lecture scenario?

The teachers made the students aware that there is not a single solution ‘but every concept map is correct in its own sense’ [line 229]. This is rather at odds with the application of experts’ maps – implying that there is a best answer to aim for. We also need to be careful about the use of expert maps. Maps produced by subject experts (but mapping novices) may not illustrate expert knowledge structures. Disciplinary experts also need some training in the application of mapping before their concept maps may be considered expert maps. Were the experts given any guidance? Was there any dialogue between them? Again, there is a paper in the making here on this point alone. What variation was there in the concept maps? What was the root concept, or root question – see work by Cañas on this. The difference between having a concept or a question to guide the map construction will cause the links in the map to be more or less dynamic.

We need to know what happens between the time points for mapping. Was there any feedback offered to the students? Research has shown that concept mapping without feedback is unlikely to have a significant effect on student learning. The feedback and the dialogue are essential aspects of the map-mediated learning process.

This leads us on to Figure 1, a map made at time point three. The quality of this map suggests the student had not received any feedback on the earlier manifestations. This is a very poor concept map for a number of reasons. There are elements that are not linked at all to the main map. There is no discernable hierarchy or clear structure so that the grammar is at odds with the grammar of the discipline. Many of the propositions are incoherent. For example, we have 'electric current / generates a magnetic field / magnetic field / you see a magnetic field'. This is a jumbled understanding that is poorly articulated in the map. There are other examples of this within the map. The problem we have here that if this exemplar is typical of the maps offered by the student, we cannot have any confidence that this is a good representation of their understanding. Following from this, any statistical analysis of the maps is just an analysis of poor quality data. It is also not clear in the example given, how are the ‘it doesn’t fit’ or the I don’t know this yet’ links applied? There seem to be no examples in figure 1.

The application of the pictograms in Table 1 is an interesting idea that could be explored much more here. The problem with their application in figure 1 is that it looks like the student simply wanted to include all 9 in the map, even if they were not really linked to the other nodes in the map. There is also another, deeper, problem with their use. By increasing the density of the information in the maps in this way, we are assuming a common understanding of the pictograms across the student cohort. There is no evidence given for this assumption. The understanding represented by the 9 pictograms needs to be unpacked and explored before they can be used in this way. Otherwise we cannot be sure we are comparing the same thing, or if students have idiosyncratic understandings underpinning each pictogram. This on its own would make an interesting paper, but it is glossed over here too superficially.

The authors have not used the literature on concept mapping to the same level as that on conceptual change. For example, we might have expected to see reference to the work by David Hay within this paper, looking at the quality of learning, and we might have expected to see some of the work on concept map morphology explored here to address research questions c and d.

The utility of the aggregated maps needs more justification. This, again, shows the difference between the needs of the researcher and those of the students involved. I cannot see how the aggregated maps will help the students here. The motivation for the study, therefore, needs to be more explicit.

Within the concept mapping literature, ‘nodes’ and ‘links’ would be the normal terminology used rather than ‘vertices’ and ‘edges’. This typical terminology should be retained.

The voices of the student are absent in this research. It would be good to hear what the students thought, rather than just use them as data points.

The English is generally good in the text, though there are a few edits required, e.g. ‘number of edges’ rather than ‘amount of edges’.

The paper needs to focus on one or two of the research questions and to proceed step-wise, considering all the points that appear to have been rushed here.

Reviewer 3 Report

Perhaps the authors, if they consider it appropriate, should explain if there are differences or there are no differences between boys and girls and if there are any and these are not significant, then indicate it equally.